# A COEFFICIENT MAKES SVRG EFFECTIVE

## ABSTRACT

Stochastic Variance Reduced Gradient (SVRG), introduced by Johnson & Zhang (2013), is a theoretically compelling optimization method. However, as Defazio & Bottou (2019) highlights, its effectiveness in deep learning is yet to be proven. In this work, we demonstrate the potential of SVRG in optimizing real-world neural networks. Our analysis finds that, for deeper networks, the strength of the variance reduction term in SVRG should be smaller and decrease as training progresses. Inspired by this, we introduce a multiplicative coefficient $\alpha$ to control the strength and adjust it through a linear decay schedule. We name our method $\alpha$-SVRG. Our results show $\alpha$-SVRG better optimizes neural networks, consistently reducing training loss compared to both baseline and the standard SVRG across various architectures and image classification datasets. We hope our findings encourage further exploration into variance reduction techniques in deep learning.

## 1 INTRODUCTION

A decade ago, Johnson & Zhang (2013) proposed a simple approach for reducing gradient variance in SGD – Stochastic Variance Reduced Gradient (SVRG). SVRG keeps a snapshot model and uses it to form a variance reduction term to adjust the gradient of the current model. This variance reduction term is the difference between the snapshot's stochastic gradient and its full gradient on the whole dataset. Utilizing this term, SVRG is able to reduce SGD's gradient variance and accelerate it to be almost as fast as the full-batch gradient descent in a strongly convex setting.

Over the years, numerous SVRG variants have emerged. Some focus on further accelerating convergence in convex settings (Xiao & Zhang, 2014; Lin et al., 2015; Defazio, 2016), while others are tailored for non-convex scenarios (Allen-Zhu & Hazan, 2016; Reddi et al., 2016; Nguyen et al., 2017; Fang et al., 2018). Both SVRG and its variants have shown effectiveness in optimizing simpler machine learning models like logistic and ridge regression (Allen-Zhu, 2017; Lei et al., 2017). In addition, SVRG has also influenced reinforcement learning (Papini et al., 2018; Du et al., 2018).

Despite the theoretical value of SVRG and its subsequent works, they have seen limited practical success in neural network training. Most SVRG research in non-convex settings is limited to modest experiments: training basic models like Multi-Layer Perceptrons (MLP) or simple CNNs on small datasets like MNIST and CIFAR-10. These studies usually exclude evaluations on more capable and deeper networks. More recently, Defazio & Bottou (2019) have exploited several variance reduction methods, including SVRG, to deep vision models. They have found that SVRG fails to reduce gradient variance for deep neural networks because the model parameters update so quickly on the loss surface that the snapshot model becomes outdated.

In this work, we show that adding a multiplicative coefficient to SVRG's variance reduction term can make it effective for deep models. Our exploration is motivated by an intriguing observation: SVRG can only reduce gradient variance in the initial training stages, but actually *increase* it later. To tackle this problem, we mathematically derive the optimal coefficient for the variance reduction term to minimize the gradient variance. Our empirical analysis then leads to two key observations about this optimal coefficient: (1) as depth of the model increases, the optimal coefficient becomes smaller; (2) as training advances, the optimal coefficient decreases, consistently dropping well below the standard SVRG's default coefficient of 1. These findings explain why a constant coefficient of 1 in standard SVRG, while initially effective, eventually fails to reduce gradient variance.

Based on these observations, we introduce a linearly decaying coefficient $\alpha$ to control the strength of the variance reduction term in SVRG. We call our method $\alpha$-SVRG and illustrate it in Figure 1.

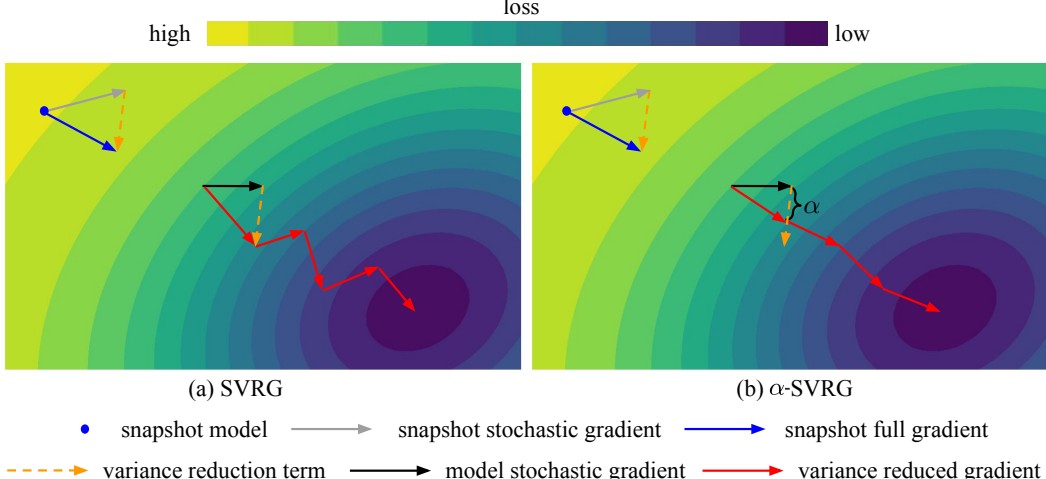

(a) SVRG            (b) $\alpha$-SVRG

- snapshot model ⟶ snapshot stochastic gradient ⟶ snapshot full gradient
⇢ variance reduction term ⟶ model stochastic gradient ⟶ variance reduced gradient

Figure 1: **SVRG vs. $\alpha$-SVRG.** Both SVRG (left) and $\alpha$-SVRG (right) use the difference between snapshot stochastic gradient (gray) and snapshot full gradient (blue) to form variance reduction term (orange), which modifies model stochastic gradient (black) into variance reduced gradient (red). But $\alpha$-SVRG employs an coefficient $\alpha$ to modulate the strength of the variance reduction term. With the coefficient, $\alpha$-SVRG reduces the variance of each update and results in faster convergence.

$\alpha$-SVRG stably reduces gradient variance and optimizes models better. We evaluate $\alpha$-SVRG on a range of architectures and multiple image classification datasets. $\alpha$-SVRG consistently achieves a lower training loss than baseline and the standard SVRG. Our results highlight the value of SVRG in deep learning. We hope our work can offer insights about SVRG and stimulate more research in variance reduction approaches for optimization in neural networks.

## 2 MOTIVATION: SVRG MAY NOT ALWAYS REDUCE VARIANCE

**SVRG formulation.** We first introduce the basic formulation of SVRG. We adopt the following notation: $t$ is the iteration index, $\theta^t$ represents the current model parameters, and $\nabla f_i(\cdot)$ denotes the gradient of loss function $f$ for the $i$-th mini-batch[1]. When the subscript $i$ is omitted, $\nabla f(\cdot)$ represents the full gradient across the entire dataset. A key concept in SVRG is the snapshot model, represented as $\widetilde{\theta}$. It is a snapshot of the model at a previous iteration before $t$, acting as an approximation of $\theta^t$. We store its full gradient $\nabla f(\widetilde{\theta})$. This snapshot is taken periodically. SVRG defines the variance reduced gradient $g_i^t$, as follows:

$$g_i^t = \nabla f_i(\theta^t) - \underbrace{(\nabla f_i(\widetilde{\theta}) - \nabla f(\widetilde{\theta}))}_{\text{variance reduction term}}.$$

(1)

Intuitively, SVRG uses the difference between the mini-batch gradient and the full gradient of a past model to modify the current mini-batch gradient. This could make $g^t$ better aligned with the full gradient and therefore stabilize each update on the current model.

Initially, SVRG was introduced in the context of vanilla SGD. Subsequent studies (Dubois-Taine et al., 2021; Wang & Klabjan, 2022) have integrated SVRG into alternative base optimizers. Following these works, we directly input the variance reduced gradient $g_i^t$ into the base optimizer. Thus, for all the experiments, SVRG and $\alpha$-SVRG use the same optimizer as the baseline. Additionally, we follow the practice in Defazio & Bottou (2019) to take the snapshot once per training epoch.

**Gradient variance.** Our goal is to assess SVRG's effectiveness in reducing gradient variance. To this end, we gather $N$ mini-batch gradients, denoted as $\{g_i^t | i \in \{1, \cdots, N\}\}$, by performing back-propagation on checkpoints of the model at the iteration $t$ with randomly selected $N$ mini-batches. For SVRG, each gradient is further modified based on Equation 1. To present a comprehensive view, we employ three existing metrics from prior studies to quantify gradient variance in Table 1.

---

[1]In SVRG's original work (Johnson & Zhang, 2013), this corresponds to the $i$-th data point.

| name | formula | description |
|---|---|---|
| metric 1* | $\frac{2}{N(N-1)} \sum_{i \neq j} \frac{1}{2} \left( 1 - \frac{\langle \boldsymbol{g}_i^t, \boldsymbol{g}_j^t \rangle}{\|\boldsymbol{g}_i^t\|^2 \|\boldsymbol{g}_j^t\|^2} \right)$ | the directional variance of the gradients |
| metric 2† | $\sum_{k=1}^{d} \mathrm{Var}(g_{i,k}^t)$ | the variance of gradients across each component |
| metric 3‡ | $\lambda_{max} \left( \frac{1}{N} \sum_{i=1}^{N} (\boldsymbol{g}_i^t - \boldsymbol{g}^t)(\boldsymbol{g}_i^t - \boldsymbol{g}^t)^T \right)$ | the magnitude of the most significant variation |

Table 1: **Gradient variance metrics.** $\boldsymbol{g}^t$ is the empirical mean of the gradients $\boldsymbol{g}_i^t$. $k$ indexes the $k$-th component of gradient $g_{i,k}^t$. References: ∗ Liu et al. (2023b), † Defazio & Bottou (2019), ‡ Jastrzebski et al. (2020).

**SVRG's effect on gradient variance.** To understand how SVRG affects training, we examine two simple models: a single linear layer (Logistic Regression) and a 4-layer Multi-Layer Perceptron (MLP-4). We train them over 30 epochs on CIFAR-10 (Krizhevsky, 2009). We compare SVRG with SGD against the baseline using only vanilla SGD.

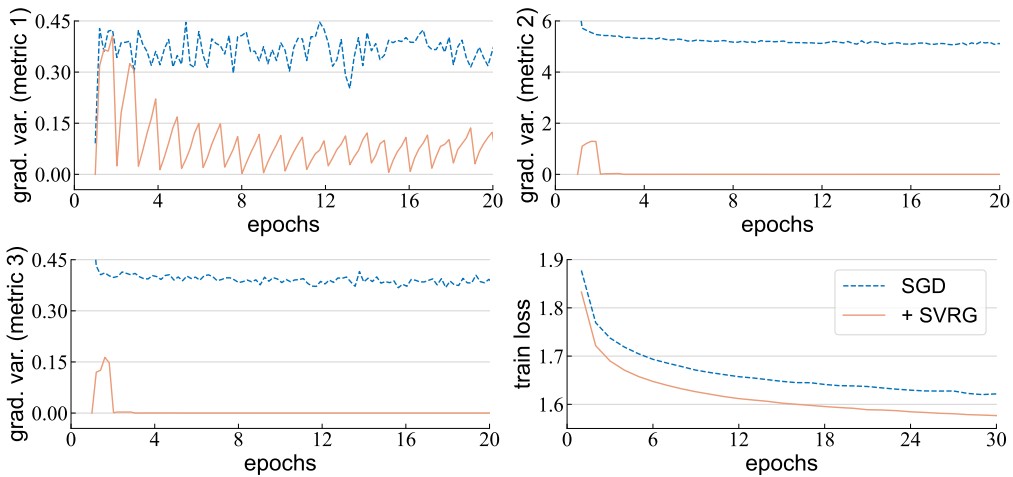

Figure 2: **SVRG on Logistic Regression.** SVRG effectively reduces the gradient variance for Logistic Regression, leading to a lower training loss than the baseline.

We plot Logistic Regression's gradient variance (top two and bottom left) and the training loss (bottom right) in Figure 2. For the Logistic Regression, SVRG consistently reduces both the gradient variance and training loss throughout the entire training process.

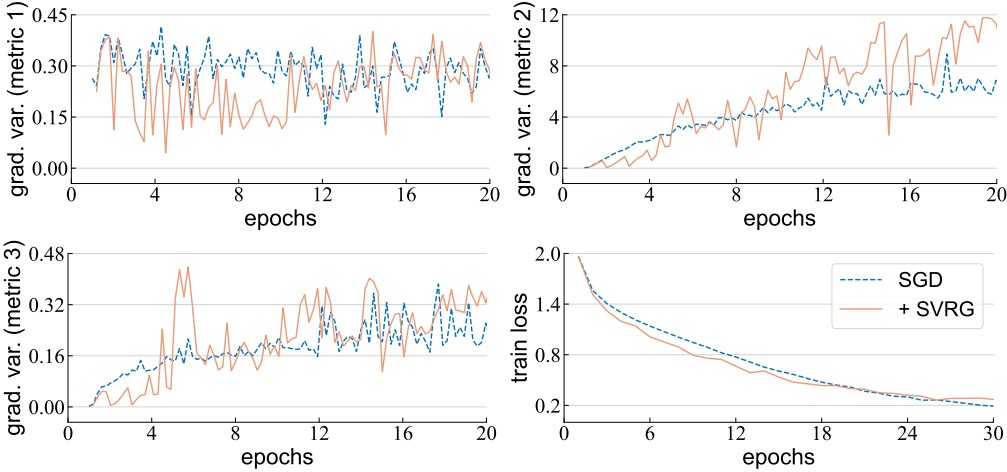

Figure 3: **SVRG on MLP-4.** In the first few epochs, SVRG substantially reduces the gradient variance for MLP-4, but afterwards, SVRG increases it, well above the baseline. As a result, SVRG exhibits a higher training loss than the baseline at the end of training.

In contrast, for MLP-4, SVRG may not always reduce gradient variance. As shown in Figure 3, SVRG only manages to decrease the gradient variance in the first five epochs. However, SVRG can greatly increase gradient variance thereafter. Consequently, SVRG has a larger final training loss than the baseline, suggesting that the increase in gradient variance hinders MLP-4's convergence.

This surprising empirical observation in a slightly deeper model leads us to question whether SVRG may alter the gradient too excessively at certain phases of training. Can we mitigate this adverse effect? We explore these questions starting from a theoretical framework in the following section.

## 3   A CLOSER LOOK AT CONTROL VARIATES IN SVRG

Control variates (Lavenberg et al., 1977) is a technique initially developed in Monte Carlo methods to reduce variance. We aim to estimate the expected value of a random variable X. The variance of this estimate usually depends on $\mathrm{Var}(X)$. To form a less variate estimate $X^*$, we can use a control variate Y that correlates with X and a coefficient $\alpha$ to regulate the influence of Y and $\mathbb{E}[Y]$ :

$$X^* = X - \alpha(Y - \mathbb{E}[Y]). \tag{2}$$

This estimate remains unbiased for any value of $\alpha$. The coefficient that minimizes the variance of the estimate can be derived as:

$$\alpha^* = \frac{\mathrm{Cov}(X, Y)}{\mathrm{Var}(Y)} = \rho(X, Y)\frac{\sigma(X)}{\sigma(Y)}, \tag{3}$$

where $\rho(X, Y)$ represents the correlation coefficient between X and Y; $\sigma(\cdot)$ denotes the standard deviation. The derivation is detailed in Appendix A. The minimized variance becomes $\mathrm{Var}(X^*) = (1 - \rho(X, Y)^2)\mathrm{Var}(X)$. The higher the correlation is, the lower the variance of the estimate is.

Note that SVRG uses control variates to reduce variance in each component of the gradient. This variance reduction occurs at each iteration $t$. Take a closer look at Equation 1 and 2: the model stochastic gradient $f_i(\boldsymbol{\theta}^t)$ is the random variable X; the snapshot stochastic gradient $f_i(\widetilde{\boldsymbol{\theta}})$ is the control variate Y; and the snapshot full gradient $f(\widetilde{\boldsymbol{\theta}})$ is the expectation $\mathbb{E}[Y]$.

A key difference between SVRG and control variates is that SVRG omits the coefficient $\alpha$, defaulting it to 1. This is possibly because the gradient distribution does not change drastically in strongly convex settings. Yet, SVRG's subsequent studies, even those addressing non-convex cases, have neglected the coefficient and formulated their theories based on Equation 1.

Motivated by this, we introduce a time-dependent coefficient vector $\boldsymbol{\alpha}^t \in \mathbb{R}^d$ in SVRG:

$$\boldsymbol{g}_i^t = \nabla f_i(\boldsymbol{\theta}^t) - \boldsymbol{\alpha}^t \odot (\nabla f_i(\widetilde{\boldsymbol{\theta}}) - \nabla f(\widetilde{\boldsymbol{\theta}})), \tag{4}$$

where $\odot$ represents the element-wise multiplication.

**Optimal coefficient.** We adopt the same gradient variance definition as Defazio & Bottou (2019) (metric 2 above) and aim to determine the optimal $\boldsymbol{\alpha}^{t*}$ that minimizes it at each iteration. Specifically, our objective is to minimize the sum of variances across each component of $\boldsymbol{g}^t$. Let $k$ index the $k$-th component $\alpha_{t,k}^*$ and the $k$-th component of the gradient $\nabla f_{\cdot,k}(\cdot)$. For clarity, we omit the mini-batch index $i$. This can be formally expressed as the following optimization problem:

$$\min_{\boldsymbol{\alpha}^t} \sum_{k=1}^{d} \mathrm{Var}(g_{\cdot,k}^t) = \sum_{k=1}^{d} \min_{\alpha_k^t} \mathrm{Var}(g_{\cdot,k}^t). \tag{5}$$

The order of minimization and summation in Equation 5 can be switched because the variance of the $k$-th component of the gradient only depends on the $k$-th component of the coefficient by our formulation. Applying Equation 3 yields the optimal coefficient $\alpha_k^{t*}$:

$$\alpha_k^{t*} = \frac{\mathrm{Cov}(\nabla f_{\cdot,k}(\widetilde{\boldsymbol{\theta}}), \nabla f_{\cdot,k}(\boldsymbol{\theta}^t))}{\mathrm{Var}(\nabla f_{\cdot,k}(\widetilde{\boldsymbol{\theta}}))} = \rho(\nabla f_{\cdot,k}(\widetilde{\boldsymbol{\theta}}), \nabla f_{\cdot,k}(\boldsymbol{\theta}^t))\frac{\sigma(\nabla f_{\cdot,k}(\boldsymbol{\theta}^t))}{\sigma(\nabla f_{\cdot,k}(\widetilde{\boldsymbol{\theta}}))}, \tag{6}$$

A stronger correlation between the snapshot and model gradients leads to a larger optimal coefficient.

For small networks like MLP-4, calculating the optimal coefficient at each iteration is feasible by gathering all mini-batch gradients for both the current and snapshot models. For larger networks, however, this method becomes impractical; we will address this challenge later in the paper.

**Observations on optimal coefficient.** To explore how the optimal coefficient evolves in a normal training setting, we train 1, 2, and 4-layer MLPs (Logistic Regression, MLP-2, and MLP-4) using SGD and AdamW (Loshchilov & Hutter, 2019) on CIFAR-10 *without using SVRG*. Given the small size of these models, we can analytically compute the optimal coefficient at each iteration. We plot its mean value over all indices $k$ in Figure 4. We can make two notable observations as below.

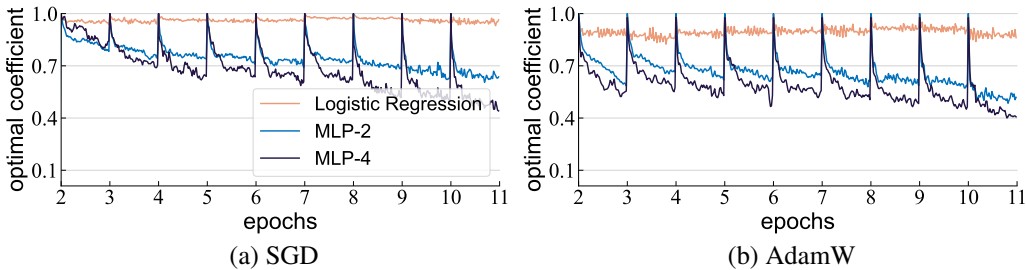

(a) SGD                    (b) AdamW

Figure 4: **Optimal coefficient.** At the start of each epoch, a snapshot is taken. Consequently, the optimal coefficient initiates at a value of 1 and results in a periodic upward jump.

*Observation 1: deeper models have smaller optimal coefficients.* For Logistic Regression, the optimal coefficient remains relatively stable, hovering near 1. For MLP-2, the coefficient deviates from 1, dropping to about 0.6. For MLP-4, it decreases more sharply, reaching approximately 0.4.

*Observation 2: average optimal coefficients of deeper model's in each epoch generally decrease as training progresses.* This suggests that as training advances, the average correlation (Equation 6) of each epoch decreases. We further analyze this epoch-wise decreasing pattern in Appendix D.

These observations shed light on why the standard SVRG struggles to reduce gradient variance or training loss in later training stages (Figure 3). A default coefficient of 1 proves to be too high and the weakening correlation between snapshot and model gradients necessitates a smaller coefficient. Without a suitable coefficient, gradient variance may increase, leading to oscillations in SGD.

**Optimal coefficient's effect on gradient variance.** We evaluate whether optimal coefficient makes SVRG more effective in reducing gradient variance. Specifically, we use SVRG with optimal coefficient to train a MLP-4 by computing the optimal coefficient (Equation 6) and adjust the gradient (Equation 4) at each iteration. We compare SVRG with optimal coefficient to the standard SVRG and the baseline. The results are presented in Figure 5. Using the optimal coefficient enables SVRG to reduce gradient variance in the early stages of training without increasing it later. The consistent gradient variance reduction yields a lower training loss than the baseline and the standard SVRG.

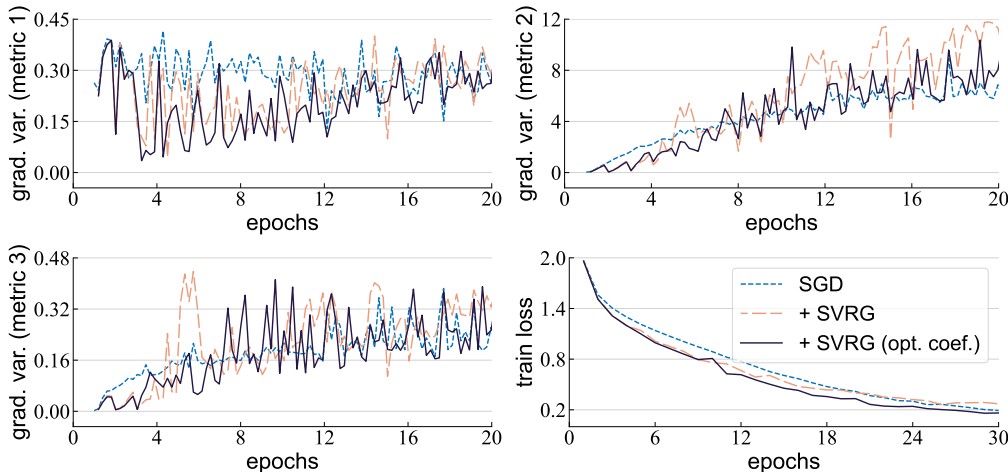

Figure 5: **SVRG with optimal coefficient on MLP-4.** SVRG with the optimal coefficient reduces gradient variance stably and achieves a substantially lower training loss than the baseline.

## 4 $\alpha$-SVRG

From our analysis above, it becomes clear that the best coefficient for SVRG is not necessarily 1 for multi-layer networks. However, computing the optimal coefficient at each iteration would result in a complexity similar to full batch gradient descent. This approach quickly becomes impractical for larger networks like ResNet (He et al., 2016). In this section, we demonstrate how using a preset schedule of $\alpha$ values can achieve a similar effect of using the computed optimal coefficients.

**$\alpha$-SVRG.** Given the decreasing trend (Figure 4) and the computational challenge, we propose to apply a linearly decreasing scalar coefficient (more in Appendix F) for SVRG, starting from an initial value $\alpha_0$ and decreasing to 0. This is our main method in this paper, and we name it $\alpha$-SVRG.

**$\alpha$-SVRG vs. optimal coefficient.** To evaluate how well $\alpha$-SVRG matches SVRG with optimal coefficient, we train a MLP-4 using $\alpha$-SVRG and compare it to SVRG with optimal coefficient and the baseline. For all experiments in this section, we set $\alpha_0 = 0.5$. The results are presented in Figure 6. Interestingly, $\alpha$-SVRG exhibits a gradient variance trend that is not much different from SVRG with optimal coefficient. A similar pattern is observed in training loss. $\alpha$-SVRG only has a slightly higher training loss than SVRG with optimal coefficient but still lower than the baseline.

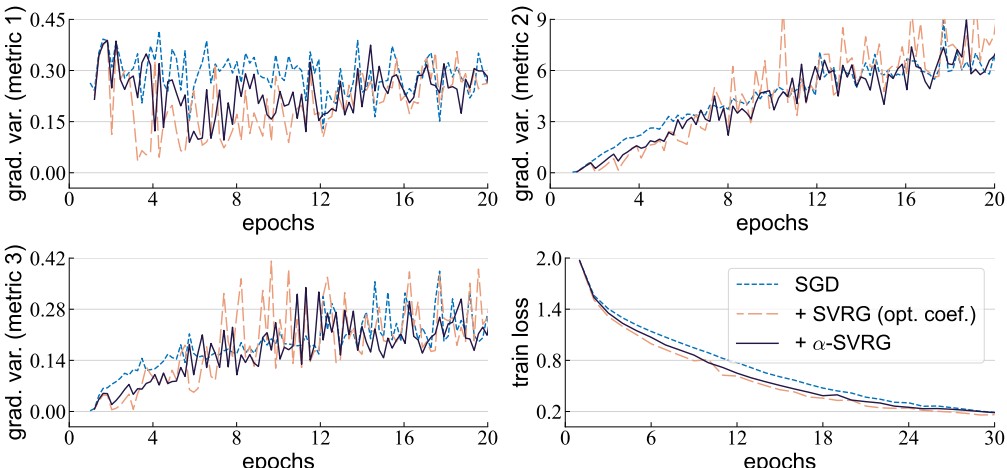

Figure 6: **$\alpha$-SVRG on MLP-4.** $\alpha$-SVRG behaves similarly to SVRG with optimal coefficient.

**$\alpha$-SVRG with AdamW.** Since AdamW (Loshchilov & Hutter, 2019) is a widely used optimizer in modern neural network training, we assess the performance of $\alpha$-SVRG with AdamW. We replace the baseline optimizer SGD with AdamW to train a MLP-4 on CIFAR-10. We compare $\alpha$-SVRG

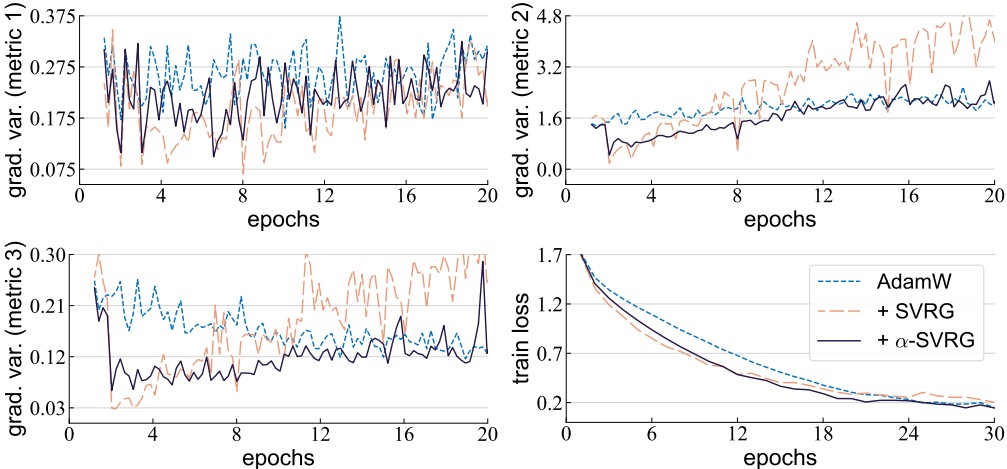

Figure 7: **$\alpha$-SVRG with AdamW on MLP-4**. $\alpha$-SVRG lowers the gradient variance throughout the entire training process, leading to a decrease in the training loss.

with the standard SVRG and the baseline. The results are shown in Figure 7. $\alpha$-SVRG exhibits a pronounced gradient variance reduction and a smaller training loss for MLP-4 than the baseline.

**$\alpha$-SVRG on deeper networks.** We further study the effectiveness of $\alpha$-SVRG with AdamW on real-world neural architectures, moving beyond simple MLPs. To this end, we train a modern ConvNet architecture, ConvNeXt-Femto (ConvNeXt-F) (Liu et al., 2022; Wightman, 2019), on CIFAR-10 using the default AdamW optimizer. We compare $\alpha$-SVRG with the standard SVRG and the baseline. We show the results in Figure 8. $\alpha$-SVRG reduces gradient variance, in contrast to the standard SVRG increasing it. Furthermore, while the training loss of the standard SVRG stagnates after around 100 epochs, the training loss of $\alpha$-SVRG continues to decrease, outperforming that of the baseline. This demonstrates the potential of $\alpha$-SVRG in handling more complex models. We further delve into this exploration with additional experiments next.

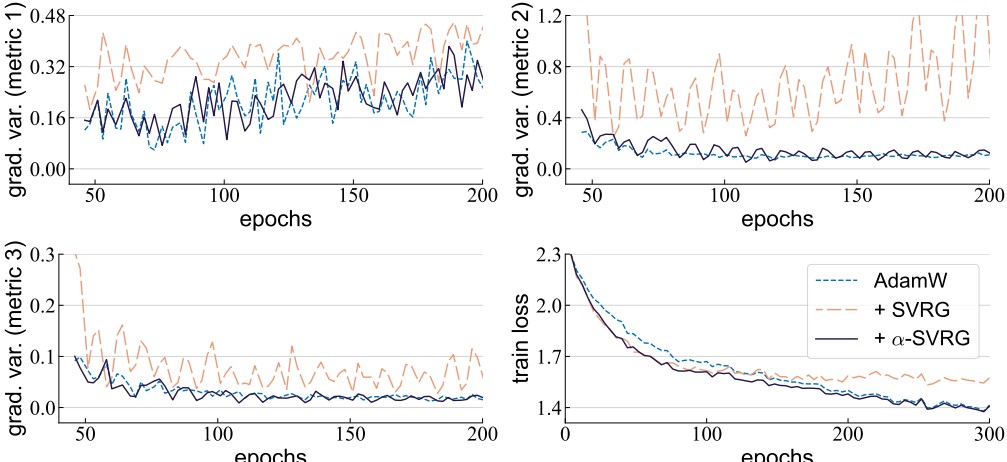

Figure 8: **$\alpha$-SVRG on ConvNeXt-F**. $\alpha$-SVRG effectively reduces the gradient variance for ConvNeXt-F, leading to faster convergence compared to the baseline.

# 5 EXPERIMENTS

## 5.1 SETTINGS

**Datasets.** We evaluate $\alpha$-SVRG using ImageNet-1K classification (Deng et al., 2009) as well as smaller image classification datasets: CIFAR-100 (Krizhevsky, 2009), Flowers (Nilsback & Zisserman, 2008), Pets (Parkhi et al., 2012), STL-10 (Coates et al., 2011), Food-101 (Bossard et al., 2014), DTD (Cimpoi et al., 2014), SVHN (Netzer et al., 2011), and EuroSAT (Helber et al., 2018; 2019).

**Models.** We use recently proposed vision models on ImageNet-1K, categorized into two groups: (1) smaller models with 5-19M parameters, including ConvNeXt-F (Wightman, 2019; Liu et al., 2022), ViT-T/16 (Dosovitskiy et al., 2021), Swin-F (Liu et al., 2021b), and Mixer-S/32 (Tolstikhin et al., 2021); (2) larger models featuring 86M and 89M parameters: ViT-B/16 and ConvNeXt-B. ConvNeXt-F is also evaluated on all smaller image classification datasets.

**Training.** We report both the training loss and the validation accuracy. Our basic training setting follows ConvNeXt (Liu et al., 2022), which uses AdamW (our baseline optimizer). On small datasets, we choose the best $\alpha_0$ from $\{0.5, 0.75, 1\}$. We find the coefficient is robust and does not require extensive tuning. Therefore, for ImageNet-1K, we set $\alpha_0$ to 0.75 for smaller models and 0.5 for larger ones. Other training settings for $\alpha$-SVRG remain the same with the baseline. Further experimental settings can be founded in Appendix B.

## 5.2 RESULTS

Table 2 presents the results of training various models on ImageNet-1K. The standard SVRG often increases the training loss, especially in larger models. In contrast, $\alpha$-SVRG decreases the training loss, confirming the effectiveness of the added coefficient to optimize real-world neural networks. This also supports our earlier finding that deeper models benefit from lower coefficient values.

| | ConvNeXt-F | | ViT-T | | Swin-F | | Mixer-S | | ViT-B | | ConvNeXt-B | |
|---|---|---|---|---|---|---|---|---|---|---|---|---|
| | train loss | | | | | | | | | | | |
| baseline | 3.487 | - | 3.443 | - | 3.427 | - | 3.635 | - | 2.817 | - | 2.644 | - |
| + SVRG | 3.505 | ↑.018 | 3.431 | ↓.012 | **3.389** | ↓.038 | 3.776 | ↑.141 | 3.309 | ↑.492 | 3.113 | ↑.469 |
| + $\alpha$-SVRG | **3.467** | ↓.020 | **3.415** | ↓.028 | 3.392 | ↓.035 | **3.609** | ↓.026 | **2.806** | ↓.011 | **2.642** | ↓.002 |
| | validation accuracy | | | | | | | | | | | |
| baseline | 76.0 | - | 73.9 | - | 74.3 | - | **71.0** | - | **81.6** | - | **83.7** | - |
| + SVRG | 75.7 | ↓0.3 | **74.3** | ↑0.4 | 74.3 | ↑0.0 | 68.8 | ↓2.2 | 78.0 | ↓3.6 | 80.8 | ↓2.9 |
| + $\alpha$-SVRG | **76.3** | ↑0.3 | 74.2 | ↑0.3 | **74.8** | ↑0.5 | 70.5 | ↓0.5 | **81.6** | ↑0.0 | 83.1 | ↓0.6 |

Table 2: **Results on ImageNet-1K.** The standard SVRG increases the train loss for most models, whereas $\alpha$-SVRG consistently decreases it for all models.

| | CIFAR-100 | | Pets | | Flowers | | STL-10 | | Food-101 | | DTD | | SVHN | | EuroSAT | |
|---|---|---|---|---|---|---|---|---|---|---|---|---|---|---|---|---|
| | train loss | | | | | | | | | | | | | | | |
| baseline | 2.66 | - | 2.20 | - | 2.40 | - | 1.64 | - | 2.45 | - | 1.98 | - | 1.59 | - | 1.25 | - |
| + SVRG | 2.94 | ↑0.28 | 3.42 | ↑1.22 | 2.26 | ↓0.14 | 1.90 | ↑0.26 | 3.03 | ↑0.58 | 2.01 | ↑0.03 | 1.64 | ↑0.05 | 1.25 | 0.00 |
| + $\alpha$-SVRG | **2.62** | ↓0.04 | **1.96** | ↓0.24 | **2.16** | ↓0.24 | **1.57** | ↓0.07 | **2.42** | ↓0.03 | **1.83** | ↓0.15 | **1.57** | ↓0.02 | **1.23** | ↓0.02 |
| | validation accuracy | | | | | | | | | | | | | | | |
| baseline | 81.0 | - | 72.8 | - | 80.8 | - | 82.3 | - | **85.9** | - | 57.9 | - | 94.9 | - | 98.1 | - |
| + SVRG | 78.2 | ↓2.8 | 17.6 | ↓55.2 | 82.6 | ↑1.8 | 65.1 | ↓17.2 | 79.6 | ↓6.3 | 57.8 | ↓0.1 | 95.7 | ↑0.8 | 97.9 | ↓0.2 |
| + $\alpha$-SVRG | **81.4** | ↑0.4 | **77.8** | ↑5.0 | **83.3** | ↑2.5 | **84.0** | ↑1.7 | **85.9** | ↑0.0 | **61.8** | ↑3.9 | **95.8** | ↑0.9 | **98.2** | ↑0.1 |

Table 3: **Results on smaller classification datasets.** While the standard SVRG mostly hurts the performance, $\alpha$-SVRG decreases the train loss and increases the validation accuracy.

Table 3 displays the results of training ConvNeXt-F on smaller image classification datasets. The standard SVRG generally elevates the training loss and impairs the generalization. On the contrary, $\alpha$-SVRG lowers the training loss and improves the validation accuracy across all small datasets.

Note that a lower training loss in $\alpha$-SVRG does not always lead to better generalization. For smaller models, a lower training loss usually directly translates to a higher validation accuracy. In larger models (Mixer-S, ViT-B, and ConvNeXt-B), additional adjustments to regularization strength may be needed for better generalization. This is out of scope for $\alpha$-SVRG as an optimization method, but warrants future research on co-adapting optimization and regularization. Intriguingly, SVRG with negative one coefficient has recently shown to be able to improve generalization (Jin et al., 2019).

## 5.3 ANALYSIS

We analyze various components from $\alpha$-SVRG. In the following experiments, we use an initial value $\alpha_0 = 0.5$ and ConvNeXt-F on STL-10 as the default setting. Because the standard SVRG is ineffective here as discussed above, we omit it and only compare $\alpha$-SVRG with an AdamW baseline.

**Coefficient value.** We investigate the impact of the initial value of the coefficient $\alpha_0$ for $\alpha$-SVRG. We vary it between 0 and 1 and observe its effect on the training loss. The results are presented in Figure 9. The favorable range for initial values in $\alpha$-SVRG is quite broad, ranging from 0.2 to 0.9. This robustness indicates $\alpha$-SVRG requires minimal tuning in the practical setting.

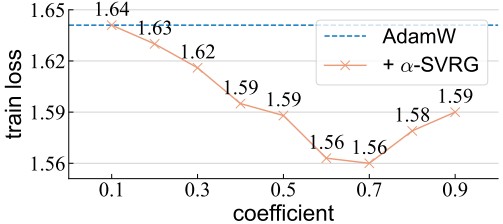

Figure 9: **Coefficient value.** $\alpha$-SVRG is effective with a wide range of coefficient values.

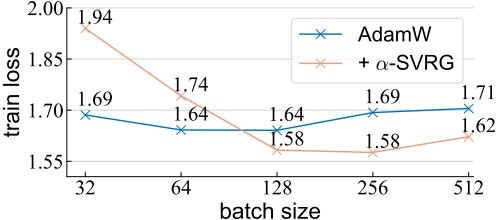

Figure 10: **Batch size.** $\alpha$-SVRG's effectiveness is observed for larger batch sizes.

**Batch size.** Since the batch size controls the variance among mini-batch data, we change the batch size to understand how it affects $\alpha$-SVRG. We also scale the learning rate linearly (Goyal et al., 2017). The default batch size is 128. In Figure 10, we can see that $\alpha$-SVRG leads to a lower training loss when the batch size is larger, but it is worse than the baseline when the batch size is smaller. This may stem from the weakening correlation between snapshot gradients and model gradients as the batch size decreases. Therefore, a sufficiently large batch size is essential for $\alpha$-SVRG.

**Inner Loop Size.** The inner loop size specifies the number of iterations between two consecutive snapshot takens. We vary it from 1 to 312 iterations to understand its effect on $\alpha$-SVRG. The default value is 39 iterations (one epoch). Figure 11 illustrates $\alpha$-SVRG has a lower training loss than the baseline even with a larger inner loop size, where snapshot is relatively distant from the current model. On the other hand, a smaller inner loop size results in a lower training loss but requires additional training time, as a full gradient must be calculated each time a snapshot is taken.

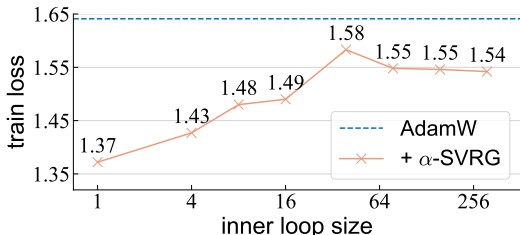

Figure 11: **Inner loop size.** Although greater inner loop size leads to weakening correlations between the model gradients and the snapshot gradients, $\alpha$-SVRG still lowers the training loss.

## 6    RELATED WORK

**Variance reduction in optimization.** There are a range of methods aiming at reducing gradient variance by directly modifying stochastic gradient. Initial works (Johnson & Zhang, 2013; Shalev-Shwartz & Zhang, 2013; Schmidt et al., 2016) focus on simple convex settings. Subsequent research further enhances these methods (Defazio et al., 2014a;b; Mairal, 2015; Babanezhad Harikandeh et al., 2015; Lin et al., 2015; Defazio, 2016; Allen-Zhu, 2017; Lin et al., 2018; Sebbouh et al., 2019) or handles finite sums in non-convex landscapes (Allen-Zhu & Hazan, 2016; Reddi et al., 2016; Nguyen et al., 2017; Lei et al., 2017; Fang et al., 2018; Li & Li, 2018; Wang et al., 2019; Elibol et al., 2020; Kavis et al., 2022). While these studies concentrate more on theoretical aspects of SVRG and do not show its effectiveness in optimizing real-world neural networks, we primarily explore the practical utility of SVRG. Gower et al. (2020) provide a comprehensive overview.

**Implicit variance reduction.** Apart from methods that explicitly adjust the gradient, there are variance reduction techniques that implicitly reduce gradient variance through other means. A variety of optimizers (Zeiler, 2012; Kingma & Ba, 2015; Dozat, 2016; Lydia & Francis, 2019; Loshchilov & Hutter, 2019; Liu et al., 2021a; 2023a; Chen et al., 2023) utilize momentum to mitigate gradient variance. They achieve this by averaging past gradients exponentially, thus stabilizing subsequent updates. Lookahead optimizer (Zhang et al., 2019) reduces gradient variance by only updating model once every $k$ iterations. Dropout (Hinton et al., 2012) is also found to reduce gradient variance and better optimize models when used at early training (Liu et al., 2023b).

## 7    CONCLUSION

Over the past decade, SVRG has been a method with significant impact in the theory of optimization. In this work, we explore the practical effectiveness of SVRG in real-world neural network training. Our key insight is the optimal strength for the variance reduction term in SVRG is not necessarily 1. It should be lower for deeper networks and decrease as training advances. This motivates us to introduce $\alpha$-SVRG: applying a linearly decreasing coefficient $\alpha$ to SVRG. $\alpha$-SVRG leads to a steady reduction in gradient variance and optimizes models better. Our experiments show that $\alpha$-SVRG consistently achieves a lower training loss compared to both the baseline and the standard SVRG. Our results motivate further research in variance reduction methods in neural network training.

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

# APPENDIX

## A DERIVATION OF THE OPTIMAL COEFFICIENT

We present the full derivation of the optimal coefficient for control variates:

$$\min_{\alpha} \text{Var}(\mathbf{X}^*) = \min_{\alpha} \text{Var}(\mathbf{X} - \alpha\mathbf{Y}) \tag{7}$$

$$= \min_{\alpha} \text{Var}(\mathbf{X}) - 2\alpha\text{Cov}(\mathbf{X}, \mathbf{Y}) + \alpha^2\text{Var}(\mathbf{Y}). \tag{8}$$

Differentiating the objective with respect to $\alpha$, we can determine the optimal coefficient $\alpha^*$:

$$2\alpha\text{Var}(\mathbf{Y}) - 2\text{Cov}(\mathbf{X}, \mathbf{Y}) = 0, \tag{9}$$

$$\implies \alpha^* = \frac{\text{Cov}(\mathbf{X}, \mathbf{Y})}{\text{Var}(\mathbf{Y})}. \tag{10}$$

Lastly, we can plug the definition of correlation coefficient:

$$\rho(\mathbf{X}, \mathbf{Y}) = \frac{\text{Cov}(\mathbf{X}, \mathbf{Y})}{\sigma(\mathbf{X})\sigma(\mathbf{Y})} \tag{11}$$

into the optimal coefficient and rewrite Equation 10 as:

$$\alpha^* = \rho(\mathbf{X}, \mathbf{Y})\frac{\sigma(\mathbf{X})}{\sigma(\mathbf{Y})}. \tag{12}$$

## B EXPERIMENTAL SETTINGS

**Training recipe.** Table 4 outlines our training recipe. It is based on the setting in ConvNeXt (Liu et al., 2022). For all experiments, the base learning rate is set at 4e-3, except for training ConvNeXt-F on ImageNet-1K using $\alpha$-SVRG, where increasing it to 8e-3 reduces the training loss very much.

| Training Setting | Configuration |
|---|---|
| weight init | trunc. normal (0.2) |
| optimizer | AdamW |
| base learning rate | 4e-3 |
| weight decay | 0.05 |
| optimizer momentum | $\beta_1, \beta_2 = 0.9, 0.999$ |
| learning rate schedule | cosine decay |
| warmup schedule | linear |
| randaugment (Cubuk et al., 2020) | (9, 0.5) |
| mixup (Zhang et al., 2018) | 0.8 |
| cutmix (Yun et al., 2019) | 1.0 |
| random erasing (Zhong et al., 2020) | 0.25 |
| label smoothing (Szegedy et al., 2016) | 0.1 |
| layer scale (Touvron et al., 2021) | 1e-6 |

Table 4: **Our basic training recipe,** adapted from ConvNeXt (Liu et al., 2022).

**Hyper-parameters.** Table 4 details the batch size, warmup epochs, and training epochs for each dataset. All the hyper-parameters selections are done on the baseline. We set the batch size in proportion to the total size of each dataset. We tune the total number of training epochs  for each

| | C-100 | Pets | Flowers | STL-10 | Food | DTD | SVHN | EuroSAT | IN1K |
|---|---|---|---|---|---|---|---|---|---|
| batch size | 1024 | 128 | 128 | 128 | 1024 | 128 | 1024 | 512 | 4096 |
| warmup epochs | 50 | 100 | 100 | 50 | 50 | 100 | 20 | 40 | 50 |
| training epochs | 300 | 600 | 600 | 300 | 300 | 600 | 100 | 200 | 300 |

Table 5: **Hyper-parameter setting.**

dataset to achieve reasonable performance with the AdamW baseline. The warmup epochs are set to one-fifth or one-sixth of the total training epochs.

We do not use stochastic depth (Huang et al., 2016) on small models. For larger models, we adhere to the original work (Dosovitskiy et al., 2021; Liu et al., 2022), using a stochastic depth rate of 0.4 for ViT-B and 0.5 for ConvNeXt-B. In these models, we maintain a consistent stochastic pattern between the current model and the snapshot at each iteration (Defazio & Bottou, 2019).

## C    DIFFERENT INITIAL COEFFICIENTS

Table 6 shows the performance of ConvNeXt-F trained with $\alpha$-SVRG using different initial co-efficients $\alpha_0$ on multiple small datasets. $\alpha$-SVRG reduces the training loss of ConvNeXt-F and enhances the validation accuracy on most datasets, regardless of the choice of initial coefficient $\alpha_0$. This demonstrates the robustness of $\alpha$-SVRG to the initial coefficient.

| | CIFAR-100 | | Pets | | Flowers | | STL-10 | |
|---|---|---|---|---|---|---|---|---|
| baseline | 2.659 | - | 2.203 | - | 2.400 | - | 1.641 | - |
| + SVRG | 2.937 | ↑0.278 | 3.424 | ↑1.221 | 2.256 | ↓0.144 | 1.899 | ↑0.258 |
| + $\alpha$-SVRG ($\alpha_0 = 0.5$) | **2.622** | ↓0.037 | 1.960 | ↓0.243 | 2.265 | ↓0.135 | 1.583 | ↓0.058 |
| + $\alpha$-SVRG ($\alpha_0 = 0.75$) | 2.646 | ↓0.013 | 2.004 | ↓0.199 | **2.162** | ↓0.238 | **1.568** | ↓0.073 |
| + $\alpha$-SVRG ($\alpha_0 = 1$) | 2.712 | ↑0.053 | **1.994** | ↓0.209 | 2.259 | ↓0.141 | 1.573 | ↓0.068 |
| | Food-101 | | DTD | | SVHN | | EuroSAT | |
| baseline | 2.451 | - | 1.980 | - | 1.588 | - | 1.247 | - |
| + SVRG | 3.026 | ↑0.575 | 2.009 | ↑0.029 | 1.639 | ↑0.051 | 1.249 | ↑0.002 |
| + $\alpha$-SVRG ($\alpha_0 = 0.5$) | **2.423** | ↓0.028 | 1.865 | ↓0.115 | **1.572** | ↓0.016 | 1.243 | ↓0.004 |
| + $\alpha$-SVRG ($\alpha_0 = 0.75$) | 2.461 | ↑0.010 | 1.829 | ↓0.151 | 1.573 | ↓0.015 | 1.237 | ↓0.010 |
| + $\alpha$-SVRG ($\alpha_0 = 1$) | 2.649 | ↑0.198 | **1.790** | ↓0.190 | 1.585 | ↓0.003 | **1.230** | ↓0.017 |

(a) **train loss**

| | CIFAR-100 | | Pets | | Flowers | | STL-10 | |
|---|---|---|---|---|---|---|---|---|
| baseline | 81.0 | - | 72.8 | - | 80.8 | - | 82.3 | - |
| + SVRG | 78.2 | ↓2.8 | 17.6 | ↓55.2 | 82.6 | ↑1.8 | 65.1 | ↓17.2 |
| + $\alpha$-SVRG ($\alpha_0 = 0.5$) | **81.4** | ↑0.4 | **77.8** | ↑5.0 | **83.3** | ↑2.5 | **83.5** | ↑1.2 |
| + $\alpha$-SVRG ($\alpha_0 = 0.75$) | 80.6 | ↓0.4 | 76.7 | ↑3.9 | 82.6 | ↑1.8 | 84.0 | ↑1.7 |
| + $\alpha$-SVRG ($\alpha_0 = 1$) | 80.0 | ↓1.0 | 77.3 | ↑4.5 | 81.9 | ↑1.1 | 84.0 | ↑1.7 |
| | Food-101 | | DTD | | SVHN | | Euro | |
| baseline | **85.9** | - | 57.9 | - | 94.9 | - | 98.1 | - |
| + SVRG | 79.6 | ↓6.3 | 57.8 | ↓0.1 | 95.7 | ↑0.8 | 97.9 | ↓0.2 |
| + $\alpha$-SVRG ($\alpha_0 = 0.5$) | **85.9** | ↑0.0 | 57.0 | ↓0.9 | 95.4 | ↑0.5 | **98.2** | ↑0.1 |
| + $\alpha$-SVRG ($\alpha_0 = 0.75$) | 85.0 | ↓0.9 | 60.3 | ↑2.4 | 95.7 | ↑0.8 | **98.2** | ↑0.1 |
| + $\alpha$-SVRG ($\alpha_0 = 1$) | 83.8 | ↓2.1 | **61.8** | ↑3.9 | **95.8** | ↑0.9 | **98.2** | ↑0.1 |

(b) **validation accuracy**

Table 6: **Results on smaller classification datasets with different initial coefficients.** While SVRG negatively affects performance on most of these datasets, $\alpha$-SVRG consistently reduces the train loss and improves the validation accuracy for almost any initial coefficient on each dataset.

## D    CORRELATION VISUALIZATION

In Equation 6, the optimal coefficient can be decomposed into a correlation and a ratio between two standard deviations. In section 3, we hypothesize it is the decreasing correlation between snapshot

gradients and model gradients that leads to the decreasing optimal coefficient. To further confirm this, we separately visualize the standard deviation ratio in Figure 12 the correlation in Figure 13.

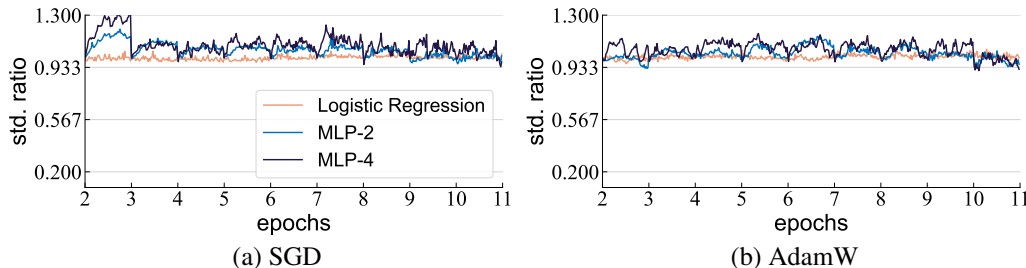

(a) SGD          (b) AdamW

Figure 12: **Standard deviation ratio.** The ratio between the standard deviations of the model gradients and the snapshot gradients sometimes oscillates around 1, but is relatively stable overall.

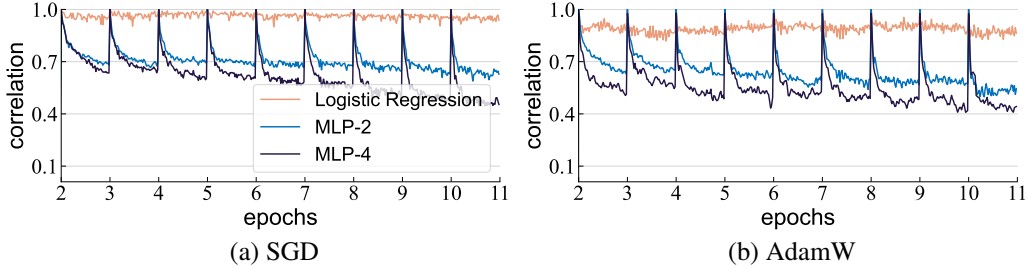

(a) SGD          (b) AdamW

Figure 13: **Correlation.** The correlation between the snapshot gradients and the model gradients behaves very similar to the optimal coefficient.

# E  STANDARD DEVIATION RESULTS

We employ 3 random seeds to run the baseline and $\alpha$-SVRG in Table 3. Table 7 presents the results. $\alpha$-SVRG consistently decreases the mean train loss and improves the mean validation accuracy. The mean difference is usually larger than one standard deviation, indicating the reliability of $\alpha$-SVRG.

|  | CIFAR-100 | Pets | Flowers | STL-10 |
|---|---|---|---|---|
| baseline | $2.645 \pm 0.013$ | $2.326 \pm 0.088$ | $2.436 \pm 0.038$ | $1.660 \pm 0.017$ |
| + $\alpha$-SVRG | $\mathbf{2.606} \pm 0.017$ | $\mathbf{2.060} \pm 0.071$ | $\mathbf{2.221} \pm 0.042$ | $\mathbf{1.577} \pm 0.022$ |
|  | Food-101 | DTD | SVHN | EuroSAT |
| baseline | $2.478 \pm 0.021$ | $2.072 \pm 0.066$ | $1.583 \pm 0.005$ | $1.259 \pm 0.017$ |
| + $\alpha$-SVRG | $\mathbf{2.426} \pm 0.007$ | $\mathbf{1.896} \pm 0.075$ | $\mathbf{1.572} \pm 0.011$ | $\mathbf{1.239} \pm 0.016$ |

(a) **train loss**

|  | CIFAR-100 | Pets | Flowers | STL-10 |
|---|---|---|---|---|
| baseline | $81.02 \pm 0.07$ | $70.61 \pm 1.55$ | $80.33 \pm 1.01$ | $80.80 \pm 1.46$ |
| + $\alpha$-SVRG | $\mathbf{81.07} \pm 0.22$ | $\mathbf{76.37} \pm 1.06$ | $\mathbf{84.15} \pm 1.15$ | $\mathbf{83.65} \pm 0.92$ |
|  | Food-101 | DTD | SVHN | EuroSAT |
| baseline | $85.29 \pm 0.47$ | $56.21 \pm 1.19$ | $94.29 \pm 0.67$ | $97.91 \pm 0.12$ |
| + $\alpha$-SVRG | $\mathbf{85.45} \pm 0.43$ | $\mathbf{61.44} \pm 0.35$ | $\mathbf{94.94} \pm 0.60$ | $\mathbf{98.13} \pm 0.07$ |

(b) **validation accuracy**

Table 7: **Results on smaller classification datasets with standard deviation.**

## F SCHEDULES

**Notations.** For clarity, we decompose the global iteration index $t$ into the epoch-wise index $s$ and the iteration index $i$ within an epoch. We also denote the total training epochs as $T$ and the number of iterations in one epoch $M$.

**Linear schedule.** Throughout the paper, we employ a coefficient that decays linearly across epochs and keeps as a constant between any two consecutive snapshots for $\alpha$-SVRG, as follows:

$$\alpha_{\text{linear}}^t = \alpha_0(1 - \frac{s}{T}), \tag{13}$$

**Other global schedules.** Nevertheless, there are other potential decaying schedules, such as quadratic decay or geometric decay. They can be formally expressed as:

$$\alpha_{\text{quadratic}}^t = \frac{\alpha_0}{T^2}s^2, \tag{14}$$

$$\alpha_{\text{geometric}}^t = \alpha_0(\frac{10^{-2}}{\alpha_0})^{\frac{s}{T}}. \tag{15}$$

**Double schedules.** In Figure 4, within an epoch, the coefficient starts from 1 and decreases over iterations. Motivated by this local behavior, we introduce three additional schedules that combine both the local and the global decrease: d(ouble)-linear, d-quadratic, and d-geometric. In addition to the global decay that schedules every epoch, each double has a local decay within each epoch that starts at 1 and decreasing to an ending value specified by the global decay.

$$\alpha_{\text{d-linear}}^t = (1 - \alpha_0(1 - \frac{s}{T}))\underbrace{(1 - \frac{i}{M})}_{\text{local decay}} + \underbrace{\alpha_0(1 - \frac{s}{T})}_{\text{global decay}} \tag{16}$$

$$\alpha_{\text{d-quadratic}}^t = (1 - \frac{\alpha_0}{T^2}s^2)\underbrace{\frac{1}{M^2}(M - i)^2}_{\text{local decay}} + \underbrace{\frac{\alpha_0}{T^2}s^2}_{\text{global decay}} \tag{17}$$

$$\alpha_{\text{d-geometric}}^t = (\alpha_0(\frac{10^{-2}}{\alpha_0})^{\frac{s}{T}} + 10^{-2})^{\frac{i}{M}} \tag{18}$$

We evaluate six above schedules with three different initial coefficients $\alpha_0$ from $\{0.5, 0.75, 1\}$ by training ConvNeXt-Femto on STL-10. Results are presented in Table 8. $\alpha$-SVRG with double schedules surprisingly increases the train loss for any initial coefficient. This could be because the locally decreasing coefficient sometimes overestimates the optimal coefficient and increases gradient variance. In contrast, $\alpha$-SVRG with global schedules consistently achieves a lower train loss than the baseline (1.641) regardless the choice of initial coefficients. This confirms our previous empirical finding that the average optimal coefficient for each epoch should decrease as training progresses.

| train loss | linear | quadratic | geometric | d-linear | d-quadratic | d-geometric |
|---|---|---|---|---|---|---|
| $\alpha_0 = 0.5$ | **1.583** | 1.607 | 1.616 | 2.067 | 1.967 | 1.808 |
| $\alpha_0 = 0.75$ | **1.568** | 1.576 | 1.582 | 2.069 | 2.003 | 1.931 |
| $\alpha_0 = 1$ | 1.573 | **1.563** | 1.574 | 1.997 | 1.970 | 1.883 |

Table 8: **Schedules.** $\alpha$-SVRG with global schedules outperforms that with double schedules.

## G DERIVATION OF THE OPTIMAL MATRIX

In Section 3, we assume that different components of the variance-reduced gradient are uncorrelated, applying Equation 3 to each component independently. In this section, we acknowledges the potential correlations between these components and extends the coefficient vector $a^t$ to the coefficient matrix $A^t$, presenting a generalized version of the optimal condition in SVRG. For notation, $A^t$ denotes the coefficient matrix; $(a_k^t)^T$ represents the row of the coefficient matrix; $a_{k,n}^t$ as the entry of the coefficient matrix at the $k$th row and $n$th column; and $K$ represents the covariance matrix.

Formally, the generalization of SVRG using the coefficient matrix can be written as:

$$\boldsymbol{g}_i^t = \nabla f_i(\boldsymbol{\theta}^t) - \boldsymbol{A}^t(\nabla f_i(\widetilde{\boldsymbol{\theta}}) - \nabla f(\widetilde{\boldsymbol{\theta}})) \tag{19}$$

We adopt the same gradient variance definition in Section3 and aim to determine the optimal coefficient matrix $\boldsymbol{A}^{t*}$ that minimizes it at each iteration.

$$\min_{\boldsymbol{A}^t} \sum_{k=1}^{d} \mathrm{Var}(g_{\cdot,k}^t) = \sum_{k=1}^{d} \min_{\boldsymbol{a}_k^t} \mathrm{Var}(g_{\cdot,k}^t) \tag{20}$$

The order of minimization and summation in Equation 5 can be switched because the variance of the $k$-th component of the gradient only depends on the $k$-th row of the coefficient matrix. We then expand the variance reduced gradient with the definition of the variance:

$$= \sum_{k=1}^{d} \min_{\boldsymbol{a}_k^t} \mathrm{Var}(\nabla f_{i,k}(\boldsymbol{\theta}^t) - (\boldsymbol{a}_k^t)^T \nabla f_i(\widetilde{\boldsymbol{\theta}})) \tag{21}$$

$$= \sum_{k=1}^{d} \min_{\boldsymbol{a}_k^t} \mathrm{Var}(\nabla f_{i,k}(\boldsymbol{\theta}^t) - \sum_{n=1}^{d} a_{k,n}^t \nabla f_{i,n}(\widetilde{\boldsymbol{\theta}})) \tag{22}$$

$$= \sum_{k=1}^{d} \min_{\boldsymbol{a}_k^t} \left( \mathrm{Var}(\nabla f_{i,k}(\boldsymbol{\theta}^t)) + \sum_{n=1}^{d} (a_{k,n}^t)^2 \mathrm{Var}(\nabla f_{i,n}(\widetilde{\boldsymbol{\theta}})) \right. \tag{23}$$

$$\left. -2 \sum_{n=1}^{d} a_{k,n}^t \mathrm{Cov}(\nabla f_{i,k}(\boldsymbol{\theta}^t), \nabla f_{i,n}(\widetilde{\boldsymbol{\theta}})) + 2 \sum_{n=1}^{d} \sum_{m \neq n} a_{k,n}^t a_{k,m}^t \mathrm{Cov}(\nabla f_{i,m}(\widetilde{\boldsymbol{\theta}}), \nabla f_{i,n}(\widetilde{\boldsymbol{\theta}})) \right) \tag{24}$$

Differentiating the objective with respect to $a_{k,n}^t$, we can determine the optimal matrix $A^{t*}$ satisfies:

$$\forall k, n : 2a_{k,n}^t \mathrm{Var}(\nabla f_{i,n}(\widetilde{\boldsymbol{\theta}})) - 2\mathrm{Cov}(\nabla f_{i,k}(\boldsymbol{\theta}^t), \nabla f_{i,n}(\widetilde{\boldsymbol{\theta}})) \tag{25}$$

$$+2 \sum_{m \neq n} \boldsymbol{a}_{k,m}^t \mathrm{Cov}(\nabla f_{i,m}(\widetilde{\boldsymbol{\theta}}), \nabla f_{i,n}(\widetilde{\boldsymbol{\theta}})) = 0 \tag{26}$$

$$\implies \forall k : \boldsymbol{K}_{\nabla f_i(\widetilde{\boldsymbol{\theta}}), \nabla f_i(\widetilde{\boldsymbol{\theta}})} \boldsymbol{a}_k^t = \begin{bmatrix} \mathrm{Cov}(\nabla f_{i,k}(\boldsymbol{\theta}^t), \nabla f_{i,1}(\widetilde{\boldsymbol{\theta}})) \\ \vdots \\ \mathrm{Cov}(\nabla f_{i,k}(\boldsymbol{\theta}^t), \nabla f_{i,d}(\widetilde{\boldsymbol{\theta}})) \end{bmatrix} \tag{27}$$

$$\implies \forall k : \boldsymbol{a}_k^t = \boldsymbol{K}_{\nabla f_i(\widetilde{\boldsymbol{\theta}}), \nabla f_i(\widetilde{\boldsymbol{\theta}})}^{-1} \begin{bmatrix} \mathrm{Cov}(\nabla f_{i,k}(\boldsymbol{\theta}^t), \nabla f_{i,1}(\widetilde{\boldsymbol{\theta}})) \\ \vdots \\ \mathrm{Cov}(\nabla f_{i,k}(\boldsymbol{\theta}^t), \nabla f_{i,d}(\widetilde{\boldsymbol{\theta}})) \end{bmatrix} \tag{28}$$

$$\implies \forall k : (\boldsymbol{a}_k^t)^T = \begin{bmatrix} \mathrm{Cov}(\nabla f_{i,k}(\boldsymbol{\theta}^t), \nabla f_{i,1}(\widetilde{\boldsymbol{\theta}})) \\ \vdots \\ \mathrm{Cov}(\nabla f_{i,k}(\boldsymbol{\theta}^t), \nabla f_{i,d}(\widetilde{\boldsymbol{\theta}})) \end{bmatrix}^T \boldsymbol{K}_{\nabla f_i(\widetilde{\boldsymbol{\theta}}), \nabla f_i(\widetilde{\boldsymbol{\theta}})}^{-1} \tag{29}$$

$$\implies \boldsymbol{A}^{t*} = \boldsymbol{K}_{\nabla f_i(\boldsymbol{\theta}^t), \nabla f_i(\widetilde{\boldsymbol{\theta}})} \boldsymbol{K}_{\nabla f_i(\widetilde{\boldsymbol{\theta}}), \nabla f_i(\widetilde{\boldsymbol{\theta}})}^{-1} \tag{30}$$

# H   STOCHASTIC RECURSIVE GRADIENTS WITH OPTIMAL COEFFICIENT

**SpiderBoost formulation.** Stochastic Recursive Gradients were invented particularly for the optimization problems in the non-convex settings (Fang et al., 2018; Nguyen et al., 2017; Wang et al., 2019; Kavis et al., 2022). In this section, we analyze one of the such methods SpiderBoost (Wang et al., 2019). Like SVRG, SpiderBoost also takes a snapshot and evaluates its full gradient $\nabla f(\widetilde{\boldsymbol{\theta}})$ periodically. Denote the iteration for taking the last snapshot as $t_0$. However, different from SVRG, it constructs the variance reduction term purely based on the information at the previous iteration. Formally, SpiderBoost defines the variance reduced gradient $\boldsymbol{g}^t$ recursively, as follows:

$$\boldsymbol{g}^t = \nabla f_i(\boldsymbol{\theta}^t) - \underbrace{(\nabla f_i(\boldsymbol{\theta}^{t-1}) - \boldsymbol{g}^{t-1})}_{\text{variance reduction term}}, \boldsymbol{g}^{t_0} = \nabla f(\widetilde{\boldsymbol{\theta}}). \tag{31}$$

**Control variates on SpiderBoost.** We can also introduce a time-dependent coefficient vector $\boldsymbol{\alpha}^t \in \mathbb{R}^d$ to control the strength of the variance reduction term in SpiderBoost:

$$\boldsymbol{g}^t = \nabla f_i(\boldsymbol{\theta}^t) - \boldsymbol{\alpha}^t \odot (\nabla f_i(\boldsymbol{\theta}^{t-1}) - \boldsymbol{g}^{t-1}), \boldsymbol{g}^{t_0} = \nabla f(\widetilde{\boldsymbol{\theta}}). \tag{32}$$

Note the previous control variates framework can no longer be applied here because the expectation of the variance reduction term is not zero. To see this, if conditioning the expectation on all the randomness from $t_0$ to $t-1$, the expectation of the first term in the variance reduction term becomes:

$$\mathbb{E}[\nabla f_i(\boldsymbol{\theta}^{t-1})] = \nabla f(\boldsymbol{\theta}^{t-1}). \tag{33}$$

However, the expectation of $g^{t-1}$ could only be the same as Equation 33 if we condition the expectation on all the randomness from $t_0$ to $t-2$.

**$\alpha$-SpiderBoost.** Despite the incompatibility of the control variates method on SpiderBoost, we still find intriguing empirical results when applying a linearly decreasing coefficient, starting from an initial value $\alpha_0$ and decreasing to 0, to SpiderBoost. We refer to this approach as $\alpha$-SpiderBoost.

We employ $\alpha$-SpiderBoost with three different initial coefficients $\alpha_0$ from $\{0.5, 0.75, 1\}$ to train ConvNeXt-Femto on a variety of small datasets. All the experiments use the same training recipe and the hyper-parameters as before (detailed in Appendix C). We present the result and compare it to the baseline, the standard SVRG, $\alpha$-SVRG, and the standard SpiderBoost in Table 9.

| | CIFAR-100 | | Pets | | Flowers | | STL-10 | |
|---|---|---|---|---|---|---|---|---|
| baseline | 2.659 | - | 2.203 | - | 2.400 | - | 1.641 | - |
| + SVRG | 2.937 | ↑0.278 | 3.424 | ↑1.221 | 2.256 | ↓0.144 | 1.899 | ↑0.258 |
| + SpiderBoost | 2.604 | ↓0.055 | **1.955** | ↓0.248 | 2.291 | ↓0.109 | **1.516** | ↓0.125 |
| + $\alpha$-SVRG ($\alpha_0 = 0.5$) | 2.622 | ↓0.037 | 1.960 | ↓0.243 | 2.265 | ↓0.135 | 1.583 | ↓0.058 |
| + $\alpha$-SVRG ($\alpha_0 = 0.75$) | 2.646 | ↓0.013 | 2.004 | ↓0.199 | **2.162** | ↓0.238 | 1.568 | ↓0.073 |
| + $\alpha$-SVRG ($\alpha_0 = 1$) | 2.712 | ↑0.053 | 1.994 | ↓0.209 | 2.259 | ↓0.141 | 1.573 | ↓0.068 |
| + $\alpha$-SpiderBoost ($\alpha_0 = 0.5$) | **2.565** | ↓0.094 | 2.032 | ↓0.171 | 2.263 | ↓0.137 | 1.544 | ↓0.097 |
| + $\alpha$-SpiderBoost ($\alpha_0 = 0.75$) | 2.567 | ↓0.092 | 1.994 | ↓0.209 | 2.266 | ↓0.134 | 1.518 | ↓0.123 |
| + $\alpha$-SpiderBoost ($\alpha_0 = 1$) | 2.585 | ↓0.074 | 1.956 | ↓0.247 | 2.265 | ↓0.135 | 1.526 | ↓0.115 |
| | Food-101 | | DTD | | SVHN | | EuroSAT | |
| baseline | 2.451 | - | 1.980 | - | 1.588 | - | 1.247 | - |
| + SVRG | 3.026 | ↑0.575 | 2.009 | ↑0.029 | 1.639 | ↑0.051 | 1.249 | ↑0.002 |
| + SpiderBoost | 2.431 | ↓0.020 | 1.978 | ↓0.002 | 1.525 | ↓0.063 | 1.289 | ↑0.042 |
| + $\alpha$-SVRG ($\alpha_0 = 0.5$) | 2.423 | ↓0.028 | 1.865 | ↓0.115 | 1.572 | ↓0.016 | 1.243 | ↓0.004 |
| + $\alpha$-SVRG ($\alpha_0 = 0.75$) | 2.461 | ↑0.010 | 1.829 | ↓0.151 | 1.573 | ↓0.015 | 1.237 | ↓0.010 |
| + $\alpha$-SVRG ($\alpha_0 = 1$) | 2.649 | ↑0.198 | **1.790** | ↓0.190 | 1.585 | ↓0.003 | **1.230** | ↓0.017 |
| + $\alpha$-SpiderBoost ($\alpha_0 = 0.5$) | **2.392** | ↓0.059 | 1.997 | ↑0.017 | **1.506** | ↓0.082 | 1.269 | ↑0.022 |
| + $\alpha$-SpiderBoost ($\alpha_0 = 0.75$) | 2.398 | ↓0.053 | 2.005 | ↑0.025 | 1.513 | ↓0.075 | 1.281 | ↑0.034 |
| + $\alpha$-SpiderBoost ($\alpha_0 = 1$) | 2.408 | ↓0.043 | 2.002 | ↑0.022 | 1.512 | ↓0.076 | 1.277 | ↑0.030 |

Table 9: $\alpha$-**SpiderBoost on smaller classification datasets.**

For small datasets (Pets, STL-10, and DTD), the standard SpiderBoost achieves a lower train loss than the baseline, but $\alpha$-SpiderBoost could have a higher train loss. Intuitively, for small datasets, the number of iterations within an epoch is small. As a result, the correlation between the model stochastic gradients and the snapshot stochastic gradients is still high enough for SpiderBoost to reduce gradient variance. This might also suggest SpiderBoost does optimize models better than SVRG on small datasets where SVRG consistently increases the train loss.

However, for relatively large datasets (CIFAR100, Food-101, and SVHN), $\alpha$-SpiderBoost could achieve a substantially lower train loss than the standard SpiderBoost and even $\alpha$-SVRG. This is likely because as the number of iterations within an epoch increases, the strength of the variance reduction term should decrease and a linearly decreasing coefficient helps weaken the strength.

The above paradoxical results of $\alpha$-SpiderBoost warrant future research to further understand how to make SpiderBoost more effective in training neural networks.

## I LIMITATION

In this work, we have shown that $\alpha$-SVRG can optimize models better by reducing gradient variance. However, we must acknowledge that it typically requires 3x computation cost compared to vanilla SGD or AdamW. In our implementation, we manage to reduce this to 2x by caching the snapshot mini-batch gradient $\nabla f_i(\theta^{\mathrm{past}})$ in memory when evaluating the snapshot full gradient $\nabla f(\theta^{\mathrm{past}})$. Later on, in constructing the variance reduced gradient, we retrieve the corresponding snapshot mini-batch gradient from memory, saving 1x computation cost. However, this approach still demands higher computational resources. Therefore, we believe it would be very valuable to explore ways to enhance $\alpha$-SVRG's efficiency in the future.

