# OpenReview forum: "A Coefficient Makes SVRG Effective"
_ICLR.cc/2024/Conference — Submitted to ICLR 2024_

### Official Review · Reviewer_auQR · 2023-10-28

**Soundness:** 3 good
**Presentation:** 3 good
**Contribution:** 2 fair
**Rating:** 5
**Confidence:** 4

**Summary:**

This paper studies SVRG for training neural networks. The main idea is introducing an additional coefficient vector $\bf\alpha$ to control the variance at each iteration, which leads to a new algorithm called $\alpha$-SVRG. The experimental results show the proposed algorithm has lower gradient variance and training loss compared to baselines.

**Strengths:**

The topic of this paper is nice. How to fill the gap between the theory of variance reduction and training neural networks has not been well-studied. This paper attempt to address this important problem.

This paper is well-organized and easy to follow. The motivation and design of $\alpha$-SVRG is clear. The authors provide sufficient numerical experiments to support their ideas.

**Weaknesses:**

I think the main weakness of this paper is its theoretical contribution is not strong.

1. Section 3 introduces ``optimal coefficient’’ by minimizing the sum of variances for each component of $\bf g^t$. However, this formulation does not consider the potential correlation between different components of $\bf g^t$. In other words, it implicitly assumes the components of $\bf g^t$ are uncorrelated, which looks too strong.

2. The optimal coefficient only considers the current variable $\theta^t$, but it is unclear how it affects the convergence rate of the algorithm in theoretical. The existing analysis only provide a greedy view, while I am more interested in the global theoretical guarantees of proposed algorithms.

3. It is well-known that stochastic recursive gradient methods has the optimal stochastic first-order oracle (SFO) complexity for nonconvex optimization. For example, SVRG can find $\epsilon$-stationary point within $n+n^{2/3}\epsilon^{-2}$ SFO calls (Allen-Zhu & Hazan, 2016; Reddi et al., 2016), while SPIDER only requires $n+n^{1/2}\epsilon^{-2}$ (Fang et al., 2018), where $n$ is the number of individual functions. Compared with SVRG, the study on SPIDER for training neural networks is more interesting.

**Questions:**

1. Can you design some strategy to improve SVRG by considering the correlation  between the component of gradient estimator?
2. Can you provide convergence analysis to show the advantage of proposed method?
3. Is it possible to apply the idea of this paper to improve stochastic recursive gradient methods?

---

> ### Author Response · Authors · 2023-11-20
> **Rebuttal by Authors**
>
> We sincerely thank you for your constructive comments. We are encouraged that you find our topic is important and our paper is easy to follow and presents sufficient numerical experiments to support $\alpha$-SVRG. We would like to address the comments and questions below.
> > Can you design some strategy to improve SVRG by considering the correlation between the components of the gradient estimator?
>
> 1. **We have provided a detailed rigorous proof that considers the correlation between the components of the gradient estimator.** If we consider the correlation, we need to add a matrix $A^t$ instead of a coefficient to control the variance reduction term in SVRG. For notation, $A^t$ denotes the coefficient matrix; $(a^t_k)^T$ represents the row of the coefficient matrix; $a^t_{k, n}$ as the entry of the coefficient matrix at the $k$-th row and $n$-th column; and $K$ represents the covariance matrix. Formally, the generalization of SVRG using the coefficient matrix can be written as:
> \begin{flalign}
>      g_i^t = \nabla f_i(\theta^t)-A^t(\nabla f_i(\widetilde{\theta})-\nabla f(\widetilde{\theta}))\end{flalign}
> We adopt the same gradient variance definition in Section3 and aim to determine the optimal coefficient matrix \(A^{t*}\) that minimizes it at each iteration.
> \begin{flalign}
>      min_{A^t} \sum_{k=1}^d Var(g^t_{\cdot, k}) =\sum_{k=1}^d min_{a^t_k} Var(g^t_{\cdot, k})\end{flalign}
> The order of minimization and summation can be switched because the variance of the $k$-th component of the gradient only depends on the $k$-th row of the coefficient matrix.
> We then expand the variance reduced gradient with the definition of the variance:
> \begin{flalign}
>      &=\sum_{k=1}^d min_{a^t_k} Var(\nabla f_{i, k}(\theta^t)-(a^t_k)^T\nabla f_i(\widetilde{\theta})) \\\\
>      &= \sum_{k=1}^d min_{a^t_k} Var(\nabla f_{i, k}(\theta^t)-\sum_{n=1}^da^t_{k, n}\nabla f_{i, n}(\widetilde{\theta})) \\\\
>      &= \sum_{k=1}^d min_{a^t_k} \bigg(Var(\nabla f_{i, k}(\theta^t)) + \sum_{n=1}^d(a^t_{k, n})^2Var(\nabla f_{i, n}(\widetilde{\theta})) - 2\sum_{n=1}^da^t_{k, n}Cov(\nabla f_{i, k}(\theta^t), \nabla f_{i, n}(\widetilde{\theta})) + 2\sum_{n=1}^d\sum_{m\neq n}a^t_{k, n}a^t_{k, m}Cov(\nabla f_{i, m}(\widetilde{\theta}), \nabla f_{i, n}(\widetilde{\theta}))\bigg)
> \end{flalign}
> Differentiating the objective with respect to $a^t_{k, n}$, we can determine the optimal matrix $A^{t*}$ satisfies:
> \begin{flalign}
>     &\forall k, n:  2a^t_{k, n}Var(\nabla f_{i, n}(\widetilde{\theta})) - 2Cov(\nabla f_{i, k}(\theta^t), \nabla f_{i, n}(\widetilde{\theta})) + 2\sum_{m\neq n}a^t_{k, m}Cov(\nabla f_{i, m}(\widetilde{\theta}), \nabla f_{i, n}(\widetilde{\theta})) = 0 \\\\      &\implies A^{t*} = K_{\nabla f_i(\theta^t), \nabla f_i(\widetilde{\theta})}K_{\nabla f_i(\widetilde{\theta}), \nabla f_i(\widetilde{\theta})}^{-1}
> \end{flalign}
> This full derivation is also included in Appendix G of the revision.
>
> 2. **Using the optimal coefficient (the diagonal entries of the optimal matrix) is a more feasible choice for empirical analysis.** For instance, a logistic regression model taking a 32x32 RGB image and outputting 10 classes has an optimal matrix with $3082^2$ entries, which cannot be easily accommodated by any modern GPU. This has similar implications in second-order optimization. Computing the Hessian matrix in deep learning is often computationally intractable, but it is possible to estimate the diagonal entries of the Hessian matrix [1], making the number of elements only scale linearly with the number of parameters in the model.

---

> ### Author Response · Authors · 2023-11-20
> **Rebuttal by Authors**
>
> > The optimal coefficient only considers the current variable $\theta^t$, but it is unclear how it affects the convergence rate of the algorithm in theoretical. The existing analysis only provides a greedy view, while I am more interested in the global theoretical guarantees of proposed algorithms. Can you provide convergence analysis to show the advantage of proposed method?
>
> Our primary aim with the development of $\alpha$-SVRG is to enhance SVRG for practical applications, particularly in training neural networks. **It's crucial to note that while theoretical analyses in SVRG offer valuable insights [2, 3], we should clarify there is a gap between theoretical analysis and empirical results in SVRG. Our work is to bridge this gap by providing a deeper understanding of why SVRG may not work in practice, and how we can adapt it to better suit real-world applications.**
>
> 1. Theoretically, SVRG has shown faster convergence than SGD in non-convex settings [2, 3]. **However, empirical studies, including our own, reveal that SVRG does not effectively reduce gradient variance or achieve lower training losses compared to SGD.** This observation holds even for simple models like MLP-4 or LeNet in non-convex settings [4].
>
> 2. Theoretically, SVRG has the same convergence rate even if the number of iterations between two consecutive snapshot scales with the number of data points [3]. **Yet, empirically, as the model moves away from the snapshot, the correlation between the snapshot gradients and the model gradients decreases, rendering the variance reduction mechanism in SVRG failed.**
>
> 3. Theoretically, the variance of the gradient estimator in SVRG is upper bounded by the iterated distance between the snapshot model and the current model up to a constant factor [2]. **But, the work [4] shows the model iterate "moves too quickly through the optimization landscape," undermining the utility of the variance upper bound.**

---

> ### Author Response · Authors · 2023-11-20
> **Rebuttal by Authors**
>
> > It is well-known that stochastic recursive gradient methods have the optimal stochastic first-order oracle (SFO) complexity for nonconvex optimization. For example, SVRG can find $\epsilon$-stationary point within $n+n^\frac{2}{3}\epsilon^{-2}$ SFO calls [2, 3], while SPIDER only requires $n+n^\frac{1}{2}\epsilon^{-2}$ [5], where $n$ is the number of individual functions. Compared with SVRG, the study on SPIDER for training neural networks is more interesting.
>
> 1. You are absolutely right about the potential of stochastic recursive gradients in training neural networks, **but we found that our current framework based on control variate cannot be applied directly to the family of stochastic recursive gradients**. The main issue is that the variance reduction term is not zero-mean in expectation. We can analyze one of the stochastic recursive gradients methods SpiderBoost as an example. Formally, SpiderBoost defines its variance reduced gradient as:
> $$g^{t} = \nabla f_i(\theta^t) - (\nabla f_i(\theta^{t-1}) - g^{t-1}), g^{t_0} = \nabla f(\widetilde{\theta}).$$
> If conditioning the expectation on all the randomness from $t_0$ to $t-1$, the expectation of the first term in the variance reduction term becomes:
> $$
>     \mathbb{E}[\nabla f_i(\theta^{t-1})] = \nabla f(\theta^{t-1}).$$
> However, the expectation of $g^{t-1}$ could only be $\nabla f(\theta^{t-1})$ if we condition the expectation on all the randomness from $t_0$ to $t-2$.
>
> 2. **Despite the incompatibility of the control variates method on SpiderBoost, we still find intriguing empirical results when applying a linearly decreasing coefficient to SpiderBoost. We refer to this approach as $\alpha$-SpiderBoost.** We employ $\alpha$-SpiderBoost with three different initial coefficients $\alpha_0$ from \{0.5, 0.75, 1\} to train ConvNeXt-Femto on a variety of small datasets. All the experiments use the same training recipe and the hyper-parameters as before (detailed in Appendix B). We present the result [here](https://drive.google.com/uc?id=1fcf9O_ZhZW25ETFEYHYT6WIIRkVukqKf) and compare it to the baseline, the standard SVRG, $\alpha$-SVRG, and the standard SpiderBoost.
> 3. **For small datasets (Pets, STL-10, and DTD), the standard SpiderBoost achieves a lower train loss than the baseline, but $\alpha$-SpiderBoost could have a higher train loss.** Intuitively, for small datasets, the number of iterations within an epoch is small. As a result, the correlation between the model stochastic gradients and the snapshot stochastic gradients is still high enough for SpiderBoost to reduce gradient variance. This might also suggest SpiderBoost does optimize models better than SVRG on small datasets where SVRG consistently increases the train loss.
> 4. **However, for relatively large datasets (CIFAR100, Food-101, and SVHN), $\alpha$-SpiderBoost could achieve a substantially lower train loss than the standard SpiderBoost and even $\alpha$-SVRG.** This is likely because as the number of iterations within an epoch increases, the strength of the variance reduction term should decrease and a linearly decreasing coefficient helps weaken the strength.
>
> The above paradoxical results of $\alpha$-SpiderBoost warrants future research to further understand how to make SpiderBoost more effective in training neural networks. **And our work on $\alpha$-SVRG could be a good starting point for this.** All the above results and analysis have been included in Appendix F.
>
> We thank you again for your valuable feedback and we hope our response can address your questions. If you have any further questions or concerns, we are very happy to answer.
>
> References:
>
> [1] Sophia: A Scalable Stochastic Second-order Optimizer for Language Model Pre-training, Anonymous, 2023.\
> [2] Variance reduction for faster non-convex optimization. Allen-Zhu & Hazan, 2016.\
> [3] Stochastic Variance Reduction for Nonconvex Optimization. Reddi et al., 2016.\
> [4] On the ineffectiveness of variance reduced optimization for deep learning. Defazio et al., 2019.\
> [5] SPIDER: Near-Optimal Non-Convex Optimization via Stochastic Path Integrated Differential Estimator. Fang, 2018.

---

### Official Review · Reviewer_LXQJ · 2023-10-29

**Soundness:** 4 excellent
**Presentation:** 3 good
**Contribution:** 2 fair
**Rating:** 5
**Confidence:** 4

**Summary:**

This paper recalls the ineffectiveness of SVRG for deep neural networks and shows that the gradients variance even increase during late epochs when training deep networks with standard SVRG.
It then introduces a modified version of SVRG that involves an $\apha$ coefficient in front of the variance reduction term.
Authors define and derive an optimal coefficient that minimizes the coordinate-wise variance of mini-batch stochastic gradients.
They show empirically that SVRG with optimal coefficient and its practical implementation $\alpha$-SVRG (linear decaying coefficient) do not suffer from increased variance.
Finally, the authors show the effectiveness of their methods compared to standard SVRG on a classification benchmark including multiple deep architectures, especially on Imagenet dataset.

**Strengths:**

- The paper clearly show the failure of SVRG for deep networks, especially at later stages of the training (confirming findings of [Defazio & Bottou, 2019])
- The motivation and derivation of section 3, the practical choice of the linear decay for alpha-SVRG emerges from clear experimental findings showing that the optimal $\alpha$ decreases in deep models
- the experiments of section 5 explore multiple scales ("smaller" and "larger" models) and families (CNN, ViT, Mixer) of deep models and show $\apha$-SVRG always lowers the training loss and often improves the test accuracy

**Weaknesses:**

- Additional parametrization (initialization, linear decay) of the proposed $\apha$-SVRG
- Results of $\apha$-SVRG often very close to the baseline: no clear improvement in most cases (except on Pets, STL-10 and DTD datasets)
- No clear explanation on how $\apha$-SVRG scales to larger datasets: no explanations about the feasibility of taking full batch gradient every epoch (cf mega-batch size of [Defazio & Bottou, 2019])
- No discussion on the supposition than the noise of SGD might be an element for better generalization (cf [Jin et al, Towards Better Generalization: BP-SVRG in Training Deep Neural Networks, 2019]

**Questions:**

**Comments**
1) $\theta^{past}$ is not a standard notation, I would recommend $\tilde{\theta}$ instead
2) An explanation of the three metrics in Table 1 is required. How are they different?
3) $g_i,k$ in metric 2 of Table 1 is not defined
4) The notations $\textbf{g}_{i \cdot}$ is unclear
5) Precising the baseline in the legends (SGD or AdamW) would be better
6) The "snapshot interval" is better known as "inner loop size"

**Questions**
1) Why are works related to alternative optimization cited page 2? No clear relevance.
2) How large are the mini-batches for $\alpha$-SVRG for the different experiments?
3) Page 5, "This is likely because the gradient direction varies more and becomes less certain as models become better trained" -> are there other works confirming this statement?
4) Page 5, is the standard deviation ratio of equation (6) constant across iterations?
5) Table 3, validation accuracy of SVRG, datasets Pets, STL-10 : have you double checked this results ? The accuracy gap is very important. How can this be explained ?
6) How does $\alpha$-SVRG behaves on Imagenet (not Imagenet-1K) ? In such a setting it is impossible to perform full gradient computations and not every epoch. cf [Defazio & Bottou, 2019]


**Suggestions**
1) Should be cited: works on optimal implementation and parameters for SVRG [Sebbouh et al. "Towards closing the gap between the theory and practice of SVRG.", 2019], [Babanezhad Harikandeh, et al. "Stopwasting my gradients: Practical svrg." 2015]
2) To enrich the bibliography, SVRG has also been extended to policy learning ([Papini et al. "Stochastic variance-reduced policy gradient." 2018] & [Du et al. "Stochastic variance reduction methods for policy evaluation." 2017]) other examples are given in [Gower, et al. "Variance-reduced methods for machine learning." 2020]

---

> ### Author Response · Authors · 2023-11-20
> **Rebuttal by Authors**
>
> We sincerely thank you for your constructive comments. We are encouraged that you find our motivation, derivations, and empirical analysis in Sections 2 and 3 are clear and the experiments are comprehensive. We would like to address the comments and questions below:
>
> > Additional parametrization (initialization, linear decay) of the proposed $\alpha$-SVRG.
>
> **We emphasize that these additional parametrization are designed to be robust, meaning they do not require extensive tuning in practical settings.** In Table 6 in Appendix B, we assess $\alpha$-SVRG across various datasets with different initial coefficients $\alpha_0 \in\\{0.5,0.75,1\\}$. The results are [here](https://drive.google.com/uc?id=1gZ46eLCA1_8CZZMMArL31lCWDF_ogfQv) and have been included in Appendix C. The consistent reduction in training loss across most datasets, irrespective of the initial coefficient choice, underscores $\alpha$-SVRG's robustness and its effectiveness.
>
> >Results of $\alpha$-SVRG often very close to the baseline: no clear improvement in most cases (except on Pets, STL-10 and DTD datasets).
>
> 1. In Appendix E, we have utilized three different random seeds to run $\alpha$-SVRG and compare it to the baseline. The results (below) demonstrate that $\alpha$-SVRG consistently decreases training loss and increases validation accuracy compared to the baseline. Importantly, the mean difference in these metrics for $\alpha$-SVRG is greater than its standard deviation in most cases. **This indicates that the improvement offered by our method is consistent.**
>
> **Train Loss**:
> |                    | CIFAR-100              | Pets                  | Flowers               | STL-10                | Food-101              | DTD                   | SVHN                  | EuroSAT               |
> |--------------------|------------------------|-----------------------|-----------------------|-----------------------|-----------------------|-----------------------|-----------------------|-----------------------|
> | baseline           | 2.645 ± 0.013          | 2.326 ± 0.088         | 2.436 ± 0.038         | 1.660 ± 0.017         | 2.478 ± 0.021         | 2.072 ± 0.066         | 1.583 ± 0.005         | 1.259 ± 0.017         |
> | + $\alpha$-SVRG $      | **2.606 ± 0.017**      | **2.060 ± 0.071**     | **2.221 ± 0.042**     | **1.577 ± 0.022**     | **2.426 ± 0.007**     | **1.896 ± 0.075**     | **1.572 ± 0.011**     | **1.239 ± 0.016**     |
>
> **Validation Accuracy:**
> |                    | CIFAR-100              | Pets                  | Flowers               | STL-10                | Food-101              | DTD                   | SVHN                  | EuroSAT               |
> |--------------------|------------------------|-----------------------|-----------------------|-----------------------|-----------------------|-----------------------|-----------------------|-----------------------|
> | baseline           | 81.02 ± 0.07           | 70.61 ± 1.55          | 80.33 ± 1.01          | 80.80 ± 1.46          | 85.29 ± 0.47          | 56.21 ± 1.19          | 94.29 ± 0.67          | 97.91 ± 0.12          |
> | + $\alpha$-SVRG $       | **81.07 ± 0.22**       | **76.37 ± 1.06**      | **84.15 ± 1.15**      | **83.65 ± 0.92**      | **85.45 ± 0.43**      | **61.44 ± 0.35**      | **94.94 ± 0.60**      | **98.13 ± 0.07**      |
>
> 2. When comparing $\alpha$-SVRG to the baseline, we maintain the same hyperparameters, including regularization strength. While $\alpha$-SVRG effectively lowers training loss, it may lead to overfitting, as indicated by the less significant improvements in validation accuracy in some datasets. This suggests a need for adjusting regularization parameters to enhance model generalization. **We believe this falls outside the scope of $\alpha$-SVRG as an optimization method.** However, it presents an important area for future research, particularly in how optimization and regularization can be co-adapted for better performance.
>
> > No clear explanation on how $\alpha$-SVRG scales to larger datasets: no explanations about the feasibility of taking full batch gradient every epoch [1]
>
> In response to your concern about the feasibility of taking full batch gradients, especially for large datasets like ImageNet21k, we emphasize that it is indeed feasible. **By partitioning the dataset into smaller mini-batches, we can compute gradients for each mini-batch independently. After computing the gradients for each mini-batch, we accumulate them to obtain the full gradient for the entire dataset.**

---

> > ### Comment · Reviewer_LXQJ · 2023-11-22
> > **Robustness wrt $alpha_0$**
> >
> > I thank the authors for taking into my remark on the additional parametrization. I checked Table 6 in the appendix and agree results seem robust wrt $alpha_0$.
> >
> > Yet, the knowledge of the number of iterations is required to perform a linear decrease to 0. This point has been omitted. This parametrization prevents the usage of the method for new problems

---

> > > ### Author Response · Authors · 2023-11-22
> > > **Response to "Robustness wrt initial coefficients"**
> > >
> > > Thank you for recognizing the robustness of the initial coefficients $\alpha_0$ in our paper.
> > >
> > > For your concern about the usage of the method for new problems, we would like to clarify that the number of iterations is a known quantity once the training recipe is established. Similarly, the decreasing schedule for the learning rate has also required the knowledge of the number of iterations to set the value at each step. But this does not prevent researchers from employing it extensively in most real-world neural network training. In this respect, we believe **our method $\alpha$-SVRG can be adapted to most neural network training problem in real-world.**

---

> > ### Comment · Reviewer_LXQJ · 2023-11-22
> > **Std and full Gradient taken for each snapshot**
> >
> > Adding Table 7 support more clearly the efficiency of the suggested method.
> >
> > I still have a concern, if the full gradient is taken at each iteration, then how aren’t there plateau on the SVRG curves when comparing to SGD? Do we agree that when entering the outer loop, the full batch Gradient is computed, equivalent to one epoch of computations? I fear the computation of the snapshot is omitted. Or are the curves smoothed?

---

> ### Author Response · Authors · 2023-11-20
> **Rebuttal by Authors**
>
> > No discussion on the supposition than the noise of SGD might be an element for better generalization [2]
>
> We acknowledge that the noise introduced by SGD can indeed contribute to better generalization in some cases. **We recognize the importance of this aspect and have included a discussion in the last paragraph of Section 5.2 in the revision:**
>
> Intriguingly, SVRG with negative one coefficient has recently shown to be able to improve generalization [2].
>
> > $\theta^{past}$ is not a standard notation, I would recommend $\widetilde{\theta}$ instead.
>
> Thank you for pointing out this notation convention. **We have updated the notation for snapshot to $\boldsymbol{\widetilde{\theta}}$ throughout the paper.**
>
>
> > An explanation of the three metrics in Table 1 is required. How are they different?
>
> **In the revision, we replace the original mathematical description with an intuitive explanation for each metric**:
> Metric 1 captures the directional variance of the gradients; metric 2 sums the variance of gradients across each component; metric 3 focuses on the magnitude of the most significant variation.
>
> > The notation $g_i$ is unclear. The notation $g_{i,k}$ in metric 2 of Table 1 is not defined.
>
> 1. **To address this, we have included the mini-batch index $i$ when formulating the variance-reduced gradient in the revision.
> $g_i^t = \nabla f_i(\theta^t)-(\nabla f_i(\widetilde{\theta})-\nabla f(\widetilde{\theta}))$.** This change has been applied throughout the paper wherever the variance-reduced gradient is mentioned, including the gradient variance paragraph at the bottom of page 2, where we describe how $g_i^t$ is collected from model checkpoints.
>
> 2. **We have added the clarification $g_{i,k}^t$ in the caption of** [Table 1](https://drive.google.com/uc?id=1dMku3LfTmax2wZZBMSzPCjXfR2ZPXn_g)**: $k$ indexes the $k$-th component of gradient $g_{i, k}^t$.**
>
> > Precising the baseline in the legends (SGD or AdamW) would be better.
>
> Thank you for pointing this out. **We have made the following improvements in the revision to ensure it is explicitly specified:**
> 1. In each plot of the revision, we have changed legend from baseline to SGD or AdamW.
> 2. Furthermore, throughout sections 2, 3, and 4, we have specified the baseline optimizer in the main text of each experiment.
> 3. In section 5, under the training paragraph of the first subsection, we have explicitly mentioned that the base optimizer employed is AdamW.
>
> > The "snapshot interval" is better known as "inner loop size"
>
> Thank you for pointing out this notation convention. **We have changed this in section 5.3 of the revision.**
>
> > Why are works related to alternative optimization cited page 2? No clear relevance.
>
> **Our original intention was to explain how SVRG can be combined with other base optimizers.** In the revised version, this paragraph has been restructured to improve readability:
>
> Initially, SVRG was introduced in the context of vanilla SGD. Subsequent studies [3,4] have integrated SVRG into alternative base optimizers. Following these works, we directly input the variance reduced gradient $g_i^t$ into the base optimizer...
>
> > How large are the mini-batches for $\alpha$-SVRG for the different experiments?
>
> **We have included all hyper-parameters, including mini-batch sizes, along with the training recipe, in Appendix B.**
> |            | C-100 | Pets | Flowers | STL-10 | Food | DTD  | SVHN | EuroSAT | ImageNet1K |
> |------------|-------|------|---------|--------|------|------|------|---------|------|
> | batch size | 1024  | 128  | 128     | 128    | 1024 | 128  | 1024 | 512     | 4096 |
>
> > Page 5, "This is likely because the gradient direction varies more and becomes less certain as models become better trained" -> are there other works confirming this statement?
>
> We realize our original message in that paragraph is not conveyed clearly. **We rephrase that paragraph as follows:**
>
> *Observation 2: average optimal coefficients of deeper model's in each epoch generally decrease as training progresses.* This suggests that as training advances, the average correlation (Equation 6) of each epoch decreases. We further analyze this epoch-wise decreasing pattern in Appendix D.
>
> > Page 5, is the standard deviation ratio of equation (6) constant across iterations?
>
> **Overall, the std ratio is relatively stable over iterations, but the correlation has a similar trend as the optimal coefficient**. The comparison is [here](https://drive.google.com/file/d/1jraqxte7uLsr_4_H1TX94mJv1fvn9Ri_/view?usp=sharing). We have also included this comparison in Appendix D.

---

> ### Author Response · Authors · 2023-11-20
> **Rebuttal by Authors**
>
> > Table 3, validation accuracy of SVRG, datasets Pets, STL-10 : have you double checked this results? The accuracy gap is very important. How can this be explained?
>
> We have double-checked the results. We have also rerun these experiments with three different seeds. The results are displayed as below:
>
> **Train Loss**:
> |                    | CIFAR-100              | Pets                  | Flowers               | STL-10                | Food-101              | DTD                   | SVHN                  | EuroSAT               |
> |--------------------|------------------------|-----------------------|-----------------------|-----------------------|-----------------------|-----------------------|-----------------------|-----------------------|
> | baseline           | 2.645 ± 0.013          | 2.326 ± 0.088         | 2.436 ± 0.038         | 1.660 ± 0.017         | 2.478 ± 0.021         | 2.072 ± 0.066         | 1.583 ± 0.005         | 1.259 ± 0.017         |
> | + $\alpha$-SVRG      | **2.606 ± 0.017**      | **2.060 ± 0.071**     | **2.221 ± 0.042**     | **1.577 ± 0.022**     | **2.426 ± 0.007**     | **1.896 ± 0.075**     | **1.572 ± 0.011**     | **1.239 ± 0.016**     |
>
> **Validation Accuracy:**
> |                    | CIFAR-100              | Pets                  | Flowers               | STL-10                | Food-101              | DTD                   | SVHN                  | EuroSAT               |
> |--------------------|------------------------|-----------------------|-----------------------|-----------------------|-----------------------|-----------------------|-----------------------|-----------------------|
> | baseline           | 81.02 ± 0.07           | 70.61 ± 1.55          | 80.33 ± 1.01          | 80.80 ± 1.46          | 85.29 ± 0.47          | 56.21 ± 1.19          | 94.29 ± 0.67          | 97.91 ± 0.12          |
> | + $\alpha$-SVRG       | **81.07 ± 0.22**       | **76.37 ± 1.06**      | **84.15 ± 1.15**      | **83.65 ± 0.92**      | **85.45 ± 0.43**      | **61.44 ± 0.35**      | **94.94 ± 0.60**      | **98.13 ± 0.07**      |
>
> 1. **The results show $\boldsymbol{\alpha}$-SVRG’s consistent decrease in train loss and improvement in validation accuracy.**
>
> 2. **We believe the significant validation accuracy improvement can be explained by the fact that the models trained on these datasets are still well within the underfitting regime.** From the values of validation accuracy, we can see they still have space to improve. Given that $\alpha$-SVRG accelerates convergence by reducing gradient variance, it can significantly impact the validation accuracy for very underfitting models.
>
> > Should be cited: works on optimal implementation and parameters for SVRG [5, 6]
>
> **In the revision, we have added these works [5, 6] into the related work section.**
>
> > To enrich the bibliography, SVRG has also been extended to policy learning [7, 8] and other examples are given in [9]
>
> **For the works [7, 8], we have added them at the end of the second paragraph in the introduction. For the work [9], we believe it is a good survey for readers to review related background information about variance reduction methods. Therefore, we add it at the first paragraph in the related works section.**
>
> We thank you again for your valuable feedback and we hope our response can address your questions. If you have any further questions or concerns, we are very happy to answer.
>
> References:
>
> [1] On the ineffectiveness of variance reduced optimization for deep learning. Defazio et al., 2019.\
> [2] Towards Better Generalization: BP-SVRG in Training Deep Neural Networks. Jin et al., 2019.\
> [3] Svrg meets adagrad: Painless variance reduction. Dubois-Taine et al., 2021.\
> [4] Divergence results and convergence of a variance reduced version of adam. Wang et al., 2022.\
> [5] Towards closing the gap between the theory and practice of SVRG. Sebbouh et al., 2019.\
> [6] Stopwasting my gradients: Practical svrg.Harikandeh, et al., 2015.\
> [7] Stochastic variance-reduced policy gradient, Papini et al., 2018.\
> [8] Stochastic variance reduction methods for policy evaluation, Du et al., 2017.\
> [9] Variance-reduced methods for machine learning. Gower, et al., 2020.

---

> > ### Comment · Reviewer_LXQJ · 2023-11-22
> > **Concern about the step size schedule, alpha-SVRG becoming SGD for efficiency and opposite conclusion with Defazio & Bottou**
> >
> > After checking again the paper, I found no reference to the fact that SVR method actually can converge using constant step size as opposed to SGD, in convex settings of course.
> >
> > This is worth mentioning, but more importantly  this changes the method. If SVRG to be efficient requires to
> > - have a decreasing alpha to zero
> > - decreasing step size
> > Then it gets very close to be SGD.
> >
> > For instance in Defazio & Bottou, the authors use piecewise constant step size.
> >
> >
> > I also have a concern about the general finding which is that alpha should go to zero at the end of the training process and thus recover SGD. This seems to be contradictory with Defazio & Bottou stating in Section 9:
> > “ As we have shown that SVRG appears to only introduce a benefit late in training,”

---

> > > ### Author Response · Authors · 2023-11-23
> > > **Response to Concern about the step size schedule, alpha-SVRG becoming SGD for efficiency and opposite conclusion with Defazio & Bottou**
> > >
> > > Thank you for pointing out these concerns. We would like to address them as follows:
> > >
> > > 1. We would like to clarify that for all initial experiments in section 2, 3, and 4 (except the one on ConvNeXt-Femto, which uses the default linear rate decay schedule), we use the same constant step size (best-tuned) for SGD (or AdamW) and SVRG. In those initial experiments, we have shown in logistic regression, SVRG can achieve lower train loss than SGD (Figure 2); in contrast, in MLP-4, SVRG will increase train loss in later stages and have higher train loss than SGD (Figure 3). Similarly, in the work [1], although the piecewise constant step size is employed, if only focusing on the first undecayed part of SGD and SVRG ([here](https://drive.google.com/uc?id=174gEujkxUDMnTvgYvRUfoyHOJmQibdom)), we can see SGD has outperformed SVRG already on deep models. **Results from our work and Defazio & Bottou[1] reinforce the idea where SGD could achieve lower train loss than SVRG with constant step size on deep non-convex models.**
> > >
> > > 2. We acknowledge using linear decay step size and coefficient does alter the original setup in SVRG. **Nevertheless, both of them are essential for SVRG to be effective in training neural networks in real-world.** The reason why SVRG can use a constant step size to converge faster than SGD in theory is because the gradient variance is upper bounded by the iterated distance between the snapshot and the current model [3, 4]. However, as Defazio & Bottou show, the iterated distance increases as the size of the model increases, rendering the variance reduction mechanism in SVRG failed for deep models. Furthermore, we find the average optimal coefficient in SVRG decreases across epoch, suggesting using a linearly decreasing coefficient is a better choice. Therefore, results from the theoretical analysis on SVRG can not be applied to $\alpha$-SVRG: $\alpha$-SVRG should not use a constant step size. We believe $\alpha$-SVRG can be viewed as a combination of vanilla SGD and the standard SVRG: it decreases the gradient variance at the early stages of training and lowers the variance reduction strength to avoid introducing additional gradient variance at the late stages of training, which reduces it eventually to SGD.
> > >
> > > 3. We acknowledge at the end of training SVRG becomes very close to vanilla SGD. **However, our method is still significantly different from SGD both in form and performance.** The linear decay schedule we employ for the coefficient lowers the coefficient proportionally across epochs. For example, if the initial coefficient starts at 1, during 50% of the training time, $\alpha$-SVRG alters the gradient at a strength higher than 0.5 and, during 90% of the training time, $\alpha$-SVRG alters the gradient at a strength higher than 0.1. This marks a clear distinction from vanilla SGD where it does not alter gradient during 100% of the training time. In addition, as Table 6 demonstrates, even if $\alpha$-SVRG with large initial coefficients, it still achieves a lower train loss than SGD.
> > >
> > > 4. We would like to clarify our method $\alpha$-SVRG does not contradict with the conclusion with Defazio & Bottou. The sentence you quoted (“As we have shown that SVRG appears to only introduce a benefit late in training,”) refers to the success of SVRG training LeNet on Cifar-10 ([here](https://drive.google.com/uc?id=1v76Jx84ds7JzrO_cb7HMwWGqW-H2Mcr-)). It shows SVRG only achieves a lower train loss than SGD at the end of training. This is likely because, as a relatively shallow network, SVRG can still help better optimize it, therefore achieving a lower train loss. In contrast, Defazio & Bottou have conducted additional fine-tuning experiments and demonstrated that SVRG can not decrease train loss even though it is used to optimize a model that has been well-trained on large datasets already. The results are [here](https://drive.google.com/uc?id=1-xhIZcIlpHd84anojCeJVNUtxFO9r6qM). As we can observe, SVRG still fails to lower the error even if it is only enabled at the later stages of training. More importantly, the lower SVRG is enabled, the closer it is to the baseline SGD. **Regardless how we choose the epoch that SVRG is enabled, it is still worse than the baseline SGD on large models.** Similarly, our work has also confirmed this idea that SVRG might be able to achieve a lower train loss on small models (Table 3) and increase the train loss on large models (Table 2).
> > >
> > > References:
> > >
> > > [1] On the ineffectiveness of variance reduced optimization for deep learning. Defazio et al., 2019.\
> > > [2] Accelerating stochastic gradient descent using predictive variance reduction. Johnson & Zhang, 2013.\
> > > [3] Variance reduction for faster non-convex optimization. Allen-Zhu & Hazan, 2016.\
> > > [4] Stochastic Variance Reduction for Nonconvex Optimization. Reddi et al., 2016.

---

> ### Author Response · Authors · 2023-11-22
> **Response to efficiency of $\alpha$-SVRG**
>
> Thank you for recognizing the effectiveness of our method.
>
> We would like to clarify that the full gradient is taken at each epoch instead of each iteration. In addition, in the training loss plot for each experiment, we did not include the train loss of the snapshot when evaluating the full gradient on the snapshot for each epoch. This convention has been adopted in the original work of SVRG [1], later works analyzing SVRG in non-convex settings [2, 3], and the work investigating the practical effectiveness of SVRG [4], when comparing the train loss of SVRG with that of SGD in experiments. **This comparison method in terms of train loss only focuses on how the train loss of the model changes after the optimizer updates the model.**
>
> References:
>
> [1] Accelerating stochastic gradient descent using predictive variance reduction. Johnson & Zhang, 2013.\
> [2] Variance reduction for faster non-convex optimization. Allen-Zhu & Hazan, 2016.\
> [3] Stochastic Variance Reduction for Nonconvex Optimization. Reddi et al., 2016.

---

### Official Review · Reviewer_3pZ5 · 2023-10-30

**Soundness:** 2 fair
**Presentation:** 2 fair
**Contribution:** 2 fair
**Rating:** 5
**Confidence:** 4

**Summary:**

The authors propose $\alpha$-SVRG, an improved version of SVRG designed to tackle the ineffectiveness of SVRG in training deep neural networks. The method involves decreasing the weight added to the variance estimated using the model snapshot. This method is obtained using

**Strengths:**

1. The paper is presented in a clear manner, especially the experiments.
2. The proposed method is effective according to the experiments.

**Weaknesses:**

1. The baseline is a bit unclear in some scenarios. See questions.
2. It seems that by saying that $\alpha_t$ decreases linearly, the authors mean that $\alpha_t=O(1/t)$. However, in some literature, a linearly decreasing sequence is a geometric sequence. The authors may want to be more clear about this.
3. Although it can be observed both intuitively and empirically that $\alpha_t$ should be decreased within an epoch, it could be a bit too arbitrary to conclude that $\alpha_t$ should be decreasing **linearly**. It's hard to see why it is preferred over other schedules, e.g., $\alpha_t=O(t^{-r})$ or $\alpha_t=O(q^{-t})$.
4. The part where the proposed method is applied to AdamW is a bit unclear. Could the authors guide me to previous works where SVRG is combined with adaptive momentum methods, if there is any? If there is, then the authors may want to refer to these works and build the proposed method based on the previous ones; otherwise it is worth discussing in greater detail how SVRG can be combined with AdamW, as the effect of the variance estimator could be more complicated due to the existence of moments.
5. The performance of the proposed method applied to ImageNet-1K is mixed compared to the baseline.

**Questions:**

1. Can the authors provide more details about the baseline? In some sections of this work, the baseline is AdamW, but I'm not sure whether this applies to all experiments.
2. Can the authors explain more about why even the  only seems to suppress variance more effectively in the early stages of training, but is less effective in later stages?

---

> ### Author Response · Authors · 2023-11-20
> **Rebuttal by Authors**
>
> We genuinely appreciate your valuable comments. We are pleased that you find our paper well-organized and acknowledge the effectiveness of $\alpha$-SVRG in training neural networks. We would like to address the comments and questions you raised:
> > The baseline is a bit unclear in some scenarios. See questions. Can the authors provide more details about the baseline? In some sections of this work, the baseline is AdamW, but I'm not sure whether this applies to all experiments.
>
> Regarding the clarity of the baseline in our experiments, **we have made improvements in the revision to ensure it is explicitly specified:**
> 1. In each plot of the revision, we have changed legend from baseline to SGD or AdamW.
> 2. Furthermore, throughout sections 2, 3, and 4, we have specified the baseline optimizer in the main text of each experiment.
> 3. In section 5, under the training paragraph of the first subsection, we have explicitly mentioned that the base optimizer employed is AdamW.
>
> > It seems that by saying that decreases linearly, the authors mean that $\alpha_t=O(1/t)$. However, in some literature, a linearly decreasing sequence is a geometric sequence. The authors may want to be more clear about this.
>
> We want to clarify that our linearly decreasing schedule for $\alpha$-SVRG is not represented as $\alpha_t=O(1/t)$. **Instead, we use a coefficient that linearly decays across epochs while remaining constant between consecutive snapshots**. For clarity, we decompose the global iteration index $t$ into the epoch-wise index $s$ and the iteration index $i$ within an epoch. We also denote the total training epochs as $T$ and the number of iterations in one epoch $M$. So the value of the default linearly decreasing coefficient at iteration $t$ can be formally written as:
> $$\alpha_\text{linear}^t = \alpha_0(1-\frac{s}{T}).$$
> Due to the limited space in the main text, we have included this formula in Appendix F of the revision and added a reference to it in the $\alpha$-SVRG paragraph in Section 4. **The reason behind choosing this schedule will also be discussed in response to your next question.**
>
> > Although it can be observed both intuitively and empirically that should be decreased within an epoch, it could be a bit too arbitrary to conclude that $\alpha_t$ should be decreasing linearly. It's hard to see why it is preferred over other schedules, e.g., $\alpha_t=O(t^{-r})$ or $\alpha_t=O(q^{-t})$
>
> 1. We agree that it is crucial to explore different scheduling methods to determine the most effective approach. **In response, we have incorporated five additional coefficient schedules into our study: quadratic, geometric, double-linear, double-quadratic, and double-geometric.** The quadratic and geometric schedules are designed to provide a global decay pattern, starting from an initial value and decreasing uniformly over epochs while remaining constant within each epoch：
> \begin{flalign}&\alpha_\text{quadratic}^t = \frac{\alpha_0}{T^2}s^2\\\\
>     &\alpha_\text{geometric}^t = \alpha_0(\frac{10^{-2}}{\alpha_0})^{\frac{s}{T}}\end{flalign}
> In contrast, the double-x schedules (double-linear, double-quadratic, and double-geometric) introduce local decays within each epoch, starting from 1 and decreasing to a value specified by the global decay pattern:
> \begin{flalign}&\alpha_\text{d-linear}^t = (1-\alpha_0(1-\frac{s}{T}))(1-\frac{i}{M}) + \alpha_0(1-\frac{s}{T})\\\\
> &\alpha_\text{d-quadratic}^t =  (1-\frac{\alpha_0}{T^2}s^2)\frac{1}{M^2}(M-i)^2+\frac{\alpha_0}{T^2}s^2\\\\
> &\alpha_\text{d-geometric}^t = (\alpha_0(\frac{10^{-2}}{\alpha_0})^{\frac{s}{T}}+10^{-2})^{\frac{i}{M}}\end{flalign}
>
> 2. We evaluated these scheduling methods by training ConvNeXt-Femto on the STL-10 dataset. The training loss results for different initial coefficients (0.5, 0.75, 1) across all six schedules are as follows:
> | train loss     | linear    | quadratic | geometric | d-linear | d-quadratic | d-geometric |
> |----------------|-----------|-----------|-----------|----------|-------------|-------------|
> | $\alpha$-SVRG $(\alpha_0=0.5)$   | **1.583** | 1.607     | 1.616     | 2.067    | 1.967       | 1.808       |
> | $\alpha$-SVRG $(\alpha_0=0.75)$  | **1.568** | 1.576     | 1.582     | 2.069    | 2.003       | 1.931       |
> | $\alpha$-SVRG $(\alpha_0=1)$     | 1.573     | **1.563** | 1.574     | 1.997    | 1.970       | 1.883       |

---

> ### Author Response · Authors · 2023-11-20
> **Rebuttal by Authors**
>
> (continue from the part 2 ...)
>
> **Our results demonstrate that the global schedules (linear, quadratic, and geometric) consistently achieved lower training losses compared to the baseline (1.641). On the other hand, the double schedules (double-linear, double-quadratic, and double-geometric) resulted in higher training losses.** This suggests that the local decreasing coefficient in the double schedules may occasionally overestimate the optimal coefficient, increasing gradient variance and hindering convergence. Conversely, the global schedules, while potentially underestimating the optimal coefficient, provide a milder and more reliable variance reduction effect, leading to consistently lower training losses. The formula of each schedule and the experiment results along with the analysis are provided in Appendix F of our revision.
>
> > The part where the proposed method is applied to AdamW is a bit unclear. Could the authors guide me to previous works where SVRG is combined with adaptive momentum methods, if there is any? If there is, then the authors may want to refer to these works and build the proposed method based on the previous ones; otherwise it is worth discussing in greater detail how SVRG can be combined with AdamW, as the effect of the variance estimator could be more complicated due to the existence of moments.
>
> **The key principle behind combining SVRG with adaptive methods, such as AdamW, is that SVRG only modifies the gradient to reduce its variance and does not update the model weights. The variance reduced gradient is then directly fed into the base optimizer to perform the weights update.** This approach ensures that the core mechanism of SVRG is independent of the baseline optimizer. These conventions are followed from previous works [1, 2]. We have included an additional paragraph on Page 2, above the gradient variance paragraph, to clarify how SVRG can be effectively combined with adaptive methods.
>
> > The performance of the proposed method applied to ImageNet-1K is mixed compared to the baseline.
>
> For smaller models, a lower training loss usually directly translates to a higher validation accuracy. However, in larger models (Mixer-S, ViT-B, and ConvNeXt-B), while $\boldsymbol{\alpha}$-SVRG lowers the training loss, it pushes the model toward the regime of overfitting, thus lowering the validation accuracy. Additional adjustments to the regularization strength may be required to achieve better generalization. **We believe that addressing the issue of overfitting and fine-tuning regularization parameters may fall outside the immediate scope of $\alpha$-SVRG as an optimization technique.** However, we recognize it is important to understand how optimization methods and regularization strategies can be co-adapted for improved model performance and generalization for future research.
>
> > Can the authors explain more about why even the only seems to suppress variance more effectively in the early stages of training, but is less effective in later stages?
>
> Assume that you are referring to SVRG. **We attribute the diminishing effectiveness of SVRG in later stages to its constant default coefficient of one.**
>
> 1. **The optimal coefficient derived in our paper reduces the variance in SVRG optimally.** The coefficient in SVRG controls the strength of the variance reduction term added to the model's stochastic gradient. This coefficient essentially balances between the unmodified stochastic gradient and a modified gradient with potentially higher variance. A coefficient of 0 means no variance reduction, whereas a very high coefficient value can lead to a significant increase in the variance of the modified gradient.
>
> 2. **The weakening correlation requires a lower optimal coefficient to effectively reduce variance without adding unwanted risk.** As illustrated in Figure 4, the optimal coefficient typically remains well below one. Therefore, using a coefficient value of one throughout the entire training process is far from optimal and poses a risk of introducing additional gradient variance. In addition, as training advances, the weakening correlation between snapshot gradients and model gradients leads to a lower optimal coefficient, further increasing the risk of introducing unwanted variance.
>
> We thank you again for your valuable feedback and we hope our response can address your questions. If you have any further questions or concerns, we are very happy to answer.
>
> References:
>
> [1] Svrg meets adagrad: Painless variance reduction. Dubois-Taine et al., 2021\
> [2] Divergence results and convergence of a variance reduced version of adam. Wang et al., 2022.

---

### Official Review · Reviewer_FPSM · 2023-11-04

**Soundness:** 3 good
**Presentation:** 3 good
**Contribution:** 3 good
**Rating:** 6
**Confidence:** 2

**Summary:**

This paper reveals that the variance reduction strength in SVRG should be lower for deep networks and decrease as training progresses.
Thus, this paper introduce a multiplicative coefficient $\alpha$ to control its
strength and adjust it with a linear decay schedule. This paper proposes a novel method  named $\alpha$-SVRG.
Experiments are conducted to  demonstrate  that $\alpha$-SVRG better optimizes neural networks, consistently
lowering the training loss compared to both baseline and standard SVRG across
various architectures and datasets. This paper is the first to bring the benefit of
SVRG into training neural networks at a practical scale.

**Strengths:**

This paper is the first to bring the benefit of
SVRG into training neural networks at a practical scale.

**Weaknesses:**

The experiments show the potential of $\alpha$-SVRG. However, the experiment results and the comparisons does not consider the computation cost. It seems that $\alpha$-SVRG takes three times computation cost as much as AdamW since $\nabla f_i(\theta^{past})$ takes extra cost and $\nabla f(\theta^{past})$ is computed for each $39$-iterations.
If  the computation cost is considered, I doubt the effciency and effectiveness of $\alpha$-SVRG.

**Questions:**

No

---

> ### Author Response · Authors · 2023-11-20
> **Rebuttal by Authors**
>
> We sincerely thank you for your comment. We appreciate your recognition of the value of $\alpha$-SVRG in training real-world neural networks. We address the question below:
> > The experiments show the potential of $\alpha$-SVRG. However, the experiment results and the comparisons do not consider the computation cost. It seems that $\alpha$-SVRG takes three times computation cost as much as AdamW since takes extra cost and is computed for each $\nabla f_i(\theta^\text{past})$ takes extra cost and $\nabla f(\theta^\text{past})$ is computed for each-39 iterations. If the computation cost is considered, I doubt the efficiency and effectiveness of $\alpha$-SVRG.
>
> 1. We acknowledge that $\alpha$-SVRG, in its basic form, appears to require a higher computational cost. **However, we have implemented an optimization that reduces this overhead from 3x to 2x compared to AdamW.** Our approach involves caching the snapshot mini-batch gradient $\nabla f_i(\theta^\text{past})$ (when evaluating the full gradient $\nabla f(\theta^\text{past})$) and utilizing it when computing the variance reduced gradient. This technique is implemented as Cache_SVRG in our codebase (supplementary material).
>
> 2. **It is important to note that the additional computational cost is a common characteristic of various variance reduction techniques, not exclusive to** $\boldsymbol{\alpha}$**-SVRG.** Similar methods [2, 3, 4, 5, 6] also exhibit increased computational demands but offer theoretical advantages over SGD. These methods all have proved in theory that the 3x increase in computation can lead to a lower error bounds compared to naively calling SGD 3x times. However, this theoretical advantage has yet to be extensively validated in practical settings.
>
> 3. **Our primary objective with $\alpha$-SVRG is to enhance its practical effectiveness.** In our experiments, $\alpha$-SVRG consistently decreases the train loss than the standard SVRG and the baseline across a variety of models and datasets. Our work on $\alpha$-SVRG is the first step towards improving the efficiency of SVRG-based methods in training real-world neural networks.
>
> We thank you again for your valuable feedback and we hope our response can address your questions. If you have any further questions or concerns, we are very happy to answer.
>
> References:
>
> [1] Saga: A fast incremental gradient method with support for non-strongly convex composite objectives. Defazio et al., 2014a.\
> [2] Katyusha: The first direct acceleration of stochastic gradient methods. Allen-Zhu, 2017.\
> [3] Spider: Near-optimal non-convex optimization via stochastic path integrated differential estimator. Fang et al., 2018.\
> [4] Sarah: A novel method for machine learning problems using stochastic recursive gradient. Nguyen et al., 2017.\
> [5] Non-convex finite-sum optimization via scsg methods. Lei et al., 2017.\
> [6] Spiderboost and momentum: Faster stochastic variance reduction algorithms. Wang et al., 2019.

---

### Author Response · Authors · 2023-11-21
**Invitation for discussions**

Dear Reviewers,

We would like to express our gratitude for your insightful feedback and suggestions, which have been very helpful in updating and enhancing our submission during the rebuttal process. We kindly invite you to review our rebuttal so that we may address any further questions you may have or clarify any points that remain unclear. In summary, our rebuttal includes the following:

- We provide results with standard deviations across three seeds and explain the significance of the consistent train loss reduction of $\alpha$-SVRG. (3pZ5 and LXQJ).

- We improve the clarity of the baseline in each experiment, including the legend of the plots and main text in section 2, 3 and 4 (3pZ5 and LXQJ).

- We separately visualize std ratio and correlation to demonstrate it is the decreasing correlation that causes the decreasing optimal coefficient (LXQJ).

- We clarify the primary goal of $\alpha$-SVRG, as an optimization method, is to reduce train loss, and whether it improves validation accuracy depends on whether the model is under the underfitting or overfitting regime.

- We ablate our default linear decay schedule against five additional schedules to reveal the advantage of our method in decreasing train loss (3pZ5).

- We combine our method with one of the stochastic recursive gradient methods SpiderBoost and present the intriguing empirical results on it (auQR).

- We develop a more generalized version of optimal SVRG, taking into account the correlation among gradient components (auQR).

- We include a limitation section to address the additional computation cost of SVRG, clarify our efforts to reduce this overhead, and suggest the value of improving the efficiency of SVRG in future works (FPSM).

- We apply a more standardized notation for SVRG throughout the paper (LXQJ).

- We add more related works about SVRG variants, its application in reinforcement learning, and potential impact on generalization (LXQJ).

We hope our responses can adequately address your concerns. We have integrated all works during the rebuttal into the revision. They are either highlighted in red if they belong to the main text or have been included as a section in Appendix from D to I. We sincerely appreciate your valuable feedback.

Best,\
Authors

---

### Meta-Review · Area_Chair_2Awz · 2023-12-13

**Metareview:**

This paper proposes an additional coefficient to improve the variance reduction of SVRG and tries to make it work for deep networks, in contrast to previous attempts.

While this direction of research is very interesting for ICLR (making variance reduction works for deep learning is still an open problem), this paper was considered quite borderline by the reviewers, and lacked support to make it acceptable in its current form to ICLR. One reviewer recommended borderline accept while the other three recommended borderline reject. The reviewers appreciated the changes made with the rebuttal, but still thought that the paper would require a revision. Some concerns that were expressed by reviewer LXQJ includes clarifying the interaction between the decreasing step-size of $\alpha$ with the size of the inner loop (one epoch here), as well as clarifying how to handle the extra cost of the batch gradient computation.

The authors are encouraged to take the detailed comments in consideration in a re-submission.

**Justification For Why Not Higher Score:**

Very borderline paper, not enough support from the reviewers.

**Justification For Why Not Lower Score:**

N/A

---

### Decision · Program_Chairs · 2024-01-16

Reject